# A FAST, WELL-FOUNDED APPROXIMATION TO THE EMPIRICAL NEURAL TANGENT KERNEL

## ABSTRACT

Empirical neural tangent kernels (eNTKs) can provide a good understanding of a given network's representation: they are often far less expensive to compute and applicable more broadly than infinite-width NTKs. For networks with $O$ output units (e.g. an $O$-class classifier), however, the eNTK on $N$ inputs is of size $NO \times NO$, taking $\mathcal{O}\left((NO)^2\right)$ memory and up to $\mathcal{O}\left((NO)^3\right)$ computation. Most existing applications have therefore used one of a handful of approximations yielding $N \times N$ kernel matrices, saving orders of magnitude of computation, but with limited to no justification. We prove that one such approximation, which we call "sum of logits," converges to the true eNTK at initialization. Our experiments demonstrate the quality of this approximation for various uses across a range of settings.

## 1 INTRODUCTION

The pursuit of a theoretical foundation for deep learning has lead researches to uncover interesting connections between neural networks (NNs) and kernel methods. It has long been known that randomly initialized NNs in the infinite width limit are Gaussian processes with what is termed the Neural Network Gaussian Process (NNGP) kernel, and training the last layer with gradient flow under squared loss corresponds to the posterior mean (Neal, 1996; Williams, 1996; Hazan & Jaakkola, 2015; Lee et al., 2017; Matthews et al., 2018; Novak et al., 2018; Yang, 2019). More recently, Jacot et al. (2018) (building off a line of closely related prior work) showed that the same is true if we train all the parameters of the network, but using a different kernel called the Neural Tangent Kernel (NTK). Yang (2020); Yang & Littwin (2021) later showed that this connection is architecturally universal, extending the domain from fully-connected NNs to most of the currently-used networks in practice, such as ResNets and Transformers. Lee et al. (2019) also showed that the dynamics of training wide but finite-width NNs with gradient descent can be approximated by a linear model obtained from the first-order Taylor expansion of that network around its initialization. Furthermore, they experimentally showed that this approximation approximation excellently holds even for networks that are not so wide.

In addition to theoretical insights from the results themselves, NTKs have had significant impact in diverse practical settings. Arora et al. (2019b) show very strong performance of NTK-based models on a variety of low-data classification and regression tasks. The condition number of an NN's NTK has been shown correlation directly with the trainability and generalization capabilities of the NN (Xiao et al., 2018; 2020); thus, Park et al. (2020); Chen et al. (2021) have used this to develop practical algorithms for neural architecture search. Wei et al. (2022); Bachmann et al. (2022) estimate the generalization ability of a specific network, randomly initialized or pre-trained on a different dataset, with efficient cross-validation. Zhou et al. (2021) use NTK regression for efficient meta-learning, and Wang et al. (2021); Holzmüller et al. (2022); Mohamadi et al. (2022) use NTKs for active learning.

There has also been significant theoretical insight gained from empirical studies of networks' NTKs. Here are a few examples: Fort et al. (2020) use NTKs to study how the loss geometry the NN evolves under gradient descent. Franceschi et al. (2021) employ NTKs to analyze the behaviour of Generative Adverserial Networks (GANs). Nguyen et al. (2020; 2021) used NTKs for dataset distillation. He et al. (2020); Adlam et al. (2020) used NTKs to predict and analyze the uncertainty of a NN's predictions. Tancik et al. (2020) use NTKs to analyze the behaviour of MLPs in learning high frequency functions, leading to new insights into our understanding of neural radiance fields.

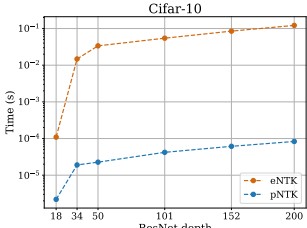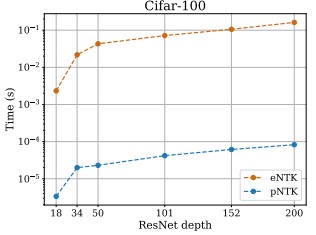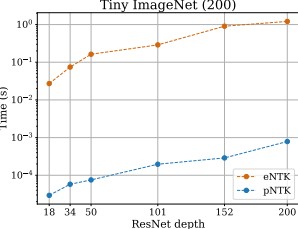

Figure 1: Comparison of **wall-clock time of evaluating the eNTK and pNTK of a pair of input datapoints** over various datasets and ResNet depths.

We thus believe NTKs will continue to be used in both theoretical and empirical deep learning.

Unfortunately, however, computing the NTK for practical networks is extremely challenging, and most of the time not even computationally feasible. The NTK of a NN is defined as the outer product of the Jacobians of the output of the NN with respect to its parameters:

$$eNTK := \Theta_\theta(x_1, x_2) = [J_\theta(f_\theta(x_1))] [J_\theta(f_\theta(x_2))]^\top , \qquad (1)$$

where $J_\theta(f_\theta(x))$ denotes the Jacobian of the function $f$ at a point $x$ with respect to the flattened vector of all its parameters, $\theta \in \mathbb{R}^P$. Assuming $f : \mathbb{R}^D \to \mathbb{R}^O$, where $D$ is the input dimension and $O$ the number of outputs, we have $J_\theta(f_\theta(x)) \in \mathbb{R}^{O \times P}$ and $\Theta_\theta(x_1, x_2) \in \mathbb{R}^{O \times O}$. Thus, computing the NTK between a set of $N_1$ data points and a set of $N_2$ data points yields $N_1 N_2$ matrices each of shape $O \times O$, which we usually reshape into an $N_1 O \times N_2 O$ matrix.

When computing an eNTK on tasks involving large datasets and with multiple output neurons, e.g. in a classification model with $O$ classes, the eNTK quickly becomes impractical regardless of how fast each entry is computed due to its $NO \times NO$ size. For instance, the full eNTK of a classification model even on the relatively mild CIFAR-10 dataset (Krizhevsky, 2009), stored in double precision, takes over 1.8 terabytes in memory. For practical usage, we need to do something better.

In this work, we present a simple trick for a strong approximation of the eNTK that removes the $O^2$ from the size of the kernel matrix, resulting in a factor of $O^2$ improvement in the memory and up to $O^3$ in computation. Since for typical classification datasets $O$ is at least 10 (e.g. CIFAR-10) and potentially $1\,000$ or more (e.g. ImageNet, Deng et al., 2009), this provides multiple orders of magnitude savings over the original eNTK (1). We prove that under appropriate initialization of the NN this approximation converges to the original eNTK at a rate of $\mathcal{O}(n^{-1/2})$ for a network of depth $L$ and width $n$ in each layer, and the predictions of kernel regression with the approximate kernel do the same. Finally, we present diverse experimental investigations supporting our theoretical results across a range of different architectures and settings. We hope this approximation further enables researches to employ NTKs towards theoretical and empirical advances in wide networks.

**Infinite NTKs** In the infinite-width limit of properly initialized NNs, $\Theta_\theta$ converges almost surely at initialization to a particular kernel, and remains constant over the course of training. Algorithms are available to compute this expectation exactly, but they tend to be substantially more expensive than computing (1) directly for all but extremely wide networks. The convergence to this infinite-width regime is also slow in practice, and moreover it eliminates some of the interest of the framework: neural architecture search, predicting generalization of a pre-trained representation, and meta-learning are all considerably less interesting when we only consider infinite-width networks that do essentially no feature learning. Thus, in this paper, we focus only on the "empirical" eNTK (1).

## 2 RELATED WORK

Among the numerous recent works that have used eNTKs either to gain insights about various phenomenons in deep learning or to propose new algorithms, not many have publicized the computational costs and implementation details of computing eNTKs. Nevertheless, all are in agreement about the expense of such computations (Park et al., 2020; Holzmüller et al., 2022; Fort et al., 2020).

Several recent works have, mostly "quietly," employed various techniques to avoid dealing with the full eNTK matrix; however, to the best of our knowledge, none provide any rigorous justifications.

Wei et al. (2022, Section 2.3) point out that if the final layer of a NN is randomly initialized, the *expected* eNTK can be written as $K_0 \otimes I_O$ for some kernel $K_0$, where $I_O$ is the $O \times O$ identity matrix and $\otimes$ is the Kronecker product. Thus, they use the approximation in which they only compute the eNTK with respect to one of the logits of the NN. Although their approach to approximating the eNTK is similar to ours, they don't provide any rigorous bounds or empirical study of how closely the actual $eNTK$ is approximated by its expectation in this regard. Wang et al. (2021) employs the same "single-logit" strategy, though they only mention the infinite-width limit as a motivation supporting their trick. Despite of these claims, we will see in our experiments that the eNTK is generally *not* diagonal. We will, however, prove upper bounds on distance of our approximation to the eNTK, and provide experimental support that this approximation captures the behaviour of the eNTK even when the NN's weights are not at initialization. Park et al. (2020); Chen et al. (2021) also seem to use a form of "single-logit" approximation to eNTK, without explicitly mentioning it. Lee et al. (2019), by contrast, do use the full eNTK, and hence never compute the kernel on more than 256 datapoints.

More recently, Novak et al. (2021) performed an in-depth analysis of computational and memory complexity required for computing the eNTK, and proposed two new approaches to reduce the time complexity of computing the eNTK in special cases (depending on the architecture of the NN) over explictly implementing (1). We remark that as our approaches are complementary, as we propose an approximation to the eNTK and Novak et al. (2021) proposes an algorithm for computation of the eNTK, which also applies when to computing our approximation – in fact, we use their "structured derivatives" method in our experiments. Moreover, their approach does not reduce the memory complexity of computing the eNTK, typically the largest burden in practical applications.

## 3   PSEUDO-NTK

We define the pseudo-NTK (pNTK) of the network $f_\theta$ as

$$pNTK \coloneqq \hat{\Theta}_\theta(x_1, x_2) = \underbrace{\left[ \nabla_\theta \left( \frac{1}{\sqrt{O}} \sum_{i=1}^{O} f_\theta^{(i)}(x_1) \right) \right]}_{1 \times P} \underbrace{\left[ \nabla_\theta \left( \frac{1}{\sqrt{O}} \sum_{i=1}^{O} f_\theta^{(i)}(x_2) \right) \right]^\top}_{P \times 1}, \quad (2)$$

where $f_\theta^{(i)}(x)$ denotes the $i$th output of $f_\theta$ on the input $x$. Unlike eNTK, which is a matrix-valued kernel for each pair of inputs, pNTK is a traditional scalar-valued kernel.

Some recent work (Arora et al., 2019a; Yang, 2020; Wei et al., 2022; Wang et al., 2021) has pointed out that in the infinite width limit the NTK $lim_{n \to \infty} \Theta(x_1, x_2)$ becomes a diagonal matrix. Thus, one can avoid computing the off-diagonal entries of the infinite-width NTK of each pair through using $\Theta_\theta(x_1, x_2) = \hat{\Theta}_\theta(x_1, x_2) \otimes I_O$, resulting in a drastic $\mathcal{O}(O^2)$ time and memory complexity decrease.

Practitioners have accordingly used the same approach in computing the corresponding eNTK of a finite width network, but with little to no further justification. We see in our experiments that for finite width networks, the NTK is **not** diagonal. In fact, we show that for most practical networks, it is very far from being diagonal, casting doubts on the validity of arguments justifying the approximation with asymptotic diagonality. We justify this category of approximation with theoretical bounds on the difference of the true NTK from the approximation (2), which we also call "sum of logits."

Before turning to the formal results and experimental evaluation, we give some intuition. First, suppose that $f_\theta^{(i)}(x) = \phi(x) \cdot v_i$, so that $v_i \in \mathbb{R}^{n_{L-1}}$ is the $i$th row of a linear read-out layer; then $\frac{1}{\sqrt{O}} \sum_{i=1}^{O} f_\theta^{(i)}(x) = \phi(x) \cdot \left[ \frac{1}{\sqrt{O}} \sum_{i=1}^{O} v_i \right]$. If the vectors $v_i \sim \mathcal{N}(0, \sigma^2 I_{n_{L-1}})$ are independent, then $\frac{1}{\sqrt{O}} \sum_{i=1}^{O} v_i \sim \mathcal{N}(0, \sigma^2 I_{n_{L-1}})$ has the same distribution as any individual entry, say $v_1$. Thus, at initialization, our sum of logits approximation agrees in distribution with the first-logit approximation. Our proof uses the sum-of-logits form, however, and we believe it may be more sensible for networks that are not at random initialization (like a pretrained network with randomized read-out layer).

Calling this vector (whether the first logit or sum of logits) $v$, we can think of (2) as the NTK of a model with a single scalar output as a function of $\phi$, whose last layer has weights $v$. When we linearize a network with that kernel for an $O$-class classification problem, getting the formula (6) discussed in Section 4.3, we end up effectively using a one-vs-rest classifier scheme. Thus, we can

think of the pseudo-NTK as approximating the process of training $O$ one-vs-rest classifiers, rather than a single $O$-way classifier.

## 4 APPROXIMATION QUALITY OF PSEUDO-NTK

We will now study various aspects of the approximation of (2) to (1), both in theory and empirically. To experimentally evaluate our claims, we present various experiments that compare the targeted characteristic of pNTK vs eNTK both at initialization and throughout training. We present our experiments on four widely-used architectures, namely, fully-connected networks (FCN) of depth 3 (as in Lee et al., 2019; 2020), fully-convolutional networks (ConvNet) of depth 8 (as in Arora et al., 2019a;b; Lee et al., 2020), ResNet18 (He et al., 2016) and WideResNet-16-$k$ (Zagoruyko & Komodakis, 2016). We evaluate each architecture at different widths, as mentioned in the plot legends: we show exact widths for FCN, while for others we show a widening factor (details in Appendix A). For consistency with most other recent papers studying NTKs and properties of NNs in general, we chose CIFAR-10 (Krizhevsky, 2009) as the dataset to evaluate our experiments on. Each experiment is repeated using three seeds, and the corresponding error bars are plotted. However, in some figures the error bars are not plotted. In those cases, it is because they interfered with clear interpretation of the plots, as the error bars were too large and not informative. We used Stochastic Gradient Descent (SGD) as the optimizer of choice for training our networks. Details on the configurations used in optimization of each network can be found in the Appendix.

Each network on each experiment is trained for 200 epochs using mini-batch stochastic gradient descent (SGD), run on NVIDIA V100 GPUs with 32GB of memory. Details on the batch sizes and learning rates used for each NN can be found in Appendix A. The measured statistic for each experiment has been reported after 0, 50, 100, 150, and 200 epochs.

### 4.1 PSEUDO-NTK CONVERGES TO ENTK AS THE NETWORK'S WIDTH GROWS

The first crucial thing to verify is whether the pNTK kernel matrix approximates the true eNTK as a whole. We study this first in terms of Frobenius norm.

**Theorem 1** (Informal). *Let $f_\theta : \mathbb{R}^D \to \mathbb{R}^O$ be a fully-connected network with layers of width $n$ whose parameters initialized according to He et al. (2015) initialization, and ReLU-type activations. Let $\hat{\Theta}_\theta(x_1, x_2)$ be the corresponding pNTK of $f_\theta$ as in (2) and $\Theta_\theta(x_1, x_2)$ the corresponding eNTK as in (1) for a fixed pair of inputs $x_1$, $x_2$. With high probability over the initialization,*

$$\frac{\|\hat{\Theta}_\theta(x_1, x_2) \otimes I_O - \Theta_\theta(x_1, x_2)\|_F}{\|\Theta_\theta(x_1, x_2)\|_F} \in \mathcal{O}(n^{-\frac{1}{2}}). \tag{3}$$

**Remark 2.** *All of the results in the paper can be straightforwardly extended to networks with different widths, as long as the consecutive layers' widths satisfy $n_{l+1} = \Theta(n_l)$. Moreover, the results can be made architecturally universal with the techniques of Yang (2020); Yang & Littwin (2021).*

Theorem 1 provides the first upper bound on the convergence rate of pNTK towards eNTK. A formal statement for Theorem 1 and its proof are in Appendix B.2.

**Remark 3.** *Based on the provided proof, it's straightforward that the ratio of information between off-diagonal and on-diagonal elements of the eNTK matrix converges to zero with a rate of $\mathcal{O}(n^{-\frac{1}{2}})$ with high probability over random initialization, as depicted in Figure 2.*

Experimental results in Figure 2 support the fact that as width grows, the ratio between off-diagonal and diagonal elements of $\Theta_\theta(x_1, x_2)$ converges to zero. Furthermore, Figure 13 provides experimental support that as width grows, $\hat{\Theta}_\theta \otimes I_O$ converges to $\Theta_\theta$ in terms of relative Frobenius norm. Note that the result of Theorem 1 only applies to epoch zero of the depicted figures, as the theorem applies only to the networks whose weights are at initialization (in the *NTK parameterization*).

However, as can be seen in the figures, these results don't necessarily apply to the NNs whose readout layers are not at initialization (i.e., after a few epochs of training). This naturally gives rise to the question: *Can the pNTK be used to analyze and represent NNs whose parameters are far from initialization?* We will now take various experimental approaches towards studying this question.

### 4.2 PSEUDO-NTK'S SPECTRUM CONVERGES TO ENTK'S AS WIDTH GROWS

As discussed before, the conditioning of a network's eNTK has been shown to be closely related to generalization properties of the network, such as trainability and generalization risk (Xiao et al., 2018; 2020; Wei et al., 2022). Thus, we would like to know how well the pNTK's eigenspectrum approximates that of the eNTK. The following theorem gives a bound on the rate of convergence between the maximum eigenvalues of the two kernels.

**Theorem 4** (Informal). *Let $f_\theta : \mathbb{R}^D \to \mathbb{R}^O$ be a fully-connected network with layers of width $n$ whose parameters initialized according to He et al. (2015) initialization, and ReLU-type activations. Let $\hat{\Theta}_\theta(x_1, x_2)$ be the corresponding pNTK of $f_\theta$ as in (2) and $\Theta_\theta(x_1, x_2)$ the corresponding eNTK as in (1) for a fixed pair of inputs $x_1, x_2$. With high probability over the initialization,*

$$\frac{\lambda_{max}\left(\hat{\Theta}_\theta(x_1, x_2) \otimes I_O\right) - \lambda_{max}\left(\Theta_\theta(x_1, x_2)\right)}{\lambda_{max}\left(\Theta_\theta(x_1, x_2)\right)} \in \mathcal{O}(n^{-\frac{1}{2}}). \tag{4}$$

Theorem 4 provides an upper bound on the difference between the maximum eigenvalue of pNTK versus the maximum eigenvalue of eNTK based on the NN's width. A formal statement for Theorem 4 and its proof are given in Appendix B.3. Experimental results in Figure 4 support the fact that as width grows, the max eigenvalue of $\Theta_\theta$ converge to the max eigenvalue of $\hat{\Theta}_\theta$. Figure 5 provides similar experimental support regarding the minimum eigenvalues of $\hat{\Theta}_\theta$ and $\Theta_\theta$ in terms of relative difference. Together, these two results intuitively imply that the condition number $\kappa(K) = \lambda_{\max}(K)/\lambda_{\min}(K)$ should become similar as width grows; this is also supported by results in Figure 6. Again, it shall be noted that Theorem 4 only applies to the NNs whose widths are at initialization and as in the previous subsection, these results don't necessarily apply to the NNs not at initialization (i.e., after a few epochs of training).

Interestingly, the rate of increase/decrease in the difference between maximum and minimum eigenvalues and the condition numbers between pNTK and eNTK do not have necessarily have a monotonic behaviour as the training goes on. Observing the exact values of $\lambda_{\min}$, $\lambda_{\max}$, and $\kappa$ for different architectures, widths at initialization and throughout training reveals that in ConvNet, WideResNet and ResNet18 architectures, $\lambda_{\min}$ is close to zero at initialization, but grows during training; the inverse phenomenon is observed with FCNs. Further investigations of these statistics might reveal interesting insights about the behaviour of NNs trained with SGD and the connections between eNTK and trainibility of the architecture.

### 4.3 KERNEL REGRESSION USING PNTK VS KERNEL REGRESSION USING ENTK

Lee et al. (2019) proved that as a finite NN's width grows, its training dynamics can be well approximated using the first-order Taylor expansion of that NN around its initialization (a *linearized neural network*). Informally, they showed that when $f$ is wide enough, its predictions after being trained using gradient descent with suitably small learning rate on the training set $\mathcal{D}$ can be approximated by those of the linearized network $f^{lin}$:

$$\underbrace{f^{lin}(x)}_{O \times 1} = \underbrace{f_0(x)}_{O \times 1} + \underbrace{\Theta_0(x, \mathcal{D})}_{O \times NO} \underbrace{\Theta_0(\mathcal{D}, \mathcal{D})^{-1}}_{NO \times NO} \underbrace{(\mathcal{Y}_\mathcal{D} - f_0(\mathcal{D}))}_{NO \times 1}, \tag{5}$$

Figure 2: Comparing the **magnitude of sum of on-diagonal and off-diagonal elements of** $\Theta_\theta$ at initialization and throughout training, based on 1000 points from CIFAR-10. The reported numbers are the average of $1000 \times 1000$ kernels each having a shape of $10 \times 10$. The same subset has then been used to train the NN using SGD. As the NN's width grows, the eNTK converges to being diagonal at initialization among all different architectures.

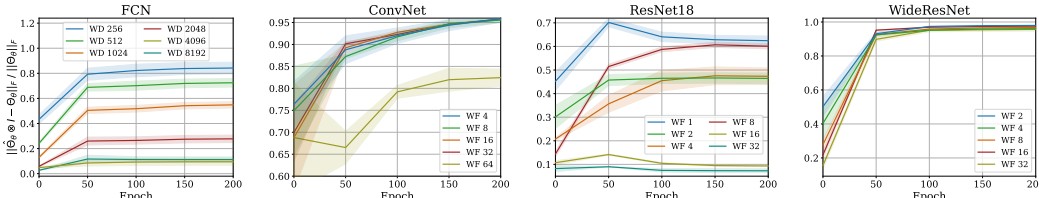

Figure 3: Evaluating the **relative difference of Frobenius norm of** $\Theta_\theta(\mathcal{D}, \mathcal{D})$ **and** $\hat{\Theta}_\theta(\mathcal{D}, \mathcal{D}) \otimes I_O$ at initialization and throughout training, based on 1000 points from CIFAR-10. As the NN's width grows, the relative difference in $\|\Theta_\theta\|_F$ and $\|\hat{\Theta}_\theta \otimes I_O\|_F$ decreases at initialization.

where $\mathcal{Y}_\mathcal{D}$ is the matrix of one-hot labels for the training points $\mathcal{D}$, and $\Theta_0$ is the eNTK of $f$ at initialization $f_0$. This is simply kernel regression on the training data $\mathcal{D}$ using the kernel $\Theta_0$ and prior mean $f_0$. Wei et al. (2022) use the same kernel in a generalized cross-validation estimator (Craven & Wahba, 1978) to predict the generalization risk of the NN. As discussed before, using the eNTK in these applications is practically infeasible, due to huge time and memory complexity of the kernel, but we show the pNTK approximates $f^{lin}(x)$ with much improved time and memory complexity.

**Theorem 5** (Informal). *Let $f_\theta : \mathbb{R}^D \to \mathbb{R}^O$ be a fully-connected network with layers of width $n$ whose parameters initialized according to He et al. (2015) initialization, and ReLU-type activations. Let $\hat{\Theta}_\theta(x_1, x_2)$ be the corresponding pNTK of $f_\theta$ as in (2) and $\Theta_\theta(x_1, x_2)$ the corresponding eNTK as in (1) for a fixed pair of inputs $x_1$, $x_2$. Define*

$$\underbrace{\hat{f}^{lin}(x)}_{1 \times O} \coloneqq \underbrace{f_0(x)}_{1 \times O} + \underbrace{\hat{\Theta}_\theta(x, \mathcal{D})}_{1 \times N} \underbrace{\hat{\Theta}_\theta(\mathcal{D}, \mathcal{D})^{-1}}_{N \times N} \underbrace{(\mathcal{Y}_\mathcal{D} - f_0(\mathcal{D}))}_{N \times O}. \tag{6}$$

*With proper reshaping, with high probability over random initialization,*

$$\|\hat{f}^{lin}(x) - f^{lin}(x)\|_F \in \mathcal{O}(n^{-\frac{1}{2}}). \tag{7}$$

Theorem 5 provides an upper bound on the norm difference of $\hat{f}^{lin}$ from (6) and $f^{lin}$ from Equation (5) based on the NN's last layer's width. A formal statement is given and proved in Appendix B.4.

Experimental results in Figure 7 show that as width grows, the predictions of kernel regression using $\hat{\Theta}_\theta$ converge to the prediction of those obtained from using $\Theta_\theta$, *while requiring orders of magnitude of less memory and time to compute*. Figure 8 shows similar results for the difference in prediction accuracies achieved using kernel regression through $\hat{\Theta}_\theta$ and $\Theta_\theta$ kernels. Appendix B.4 also shows further analysis of how well the *linearized network* predicts the final accuracy of the trained model for each architecture and width pair. Although $\|\hat{f}^{lin}(x) - f^{lin}(x)\|_F$ decreases with width of the network in Figure 7 at initialization, this does not *necessarily* translate to a monotonic behaviour in prediction accuracies, a non-smooth function of the vector of predictions; we do see that the expected pattern more or less holds, however.

A surprising outcome depicted in Figures 7 and 8 is that while training the model's parameters, predictions of $\hat{f}^{lin}$ and $f^{lin}$ converge very quickly. This is particularly intriguing as it's in contrast

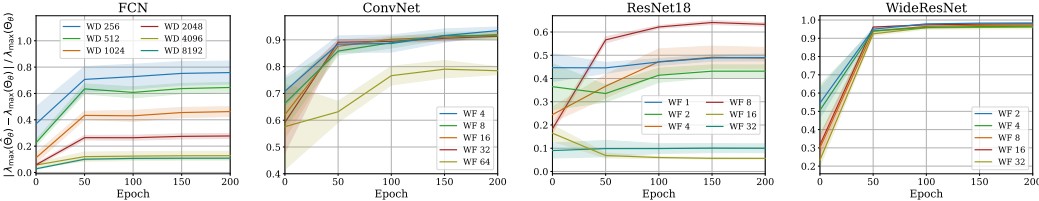

Figure 4: Evaluating the **relative difference of** $\lambda_{\max}$ **of** $\Theta_\theta(\mathcal{D}, \mathcal{D})$ **and** $\hat{\Theta}_\theta(\mathcal{D}, \mathcal{D})$ at initialization and throughout training, based on kernels on a subset ($|\mathcal{D}| = 1000$) of points from CIFAR-10. As the NN's width grows, the relative difference in $\lambda_{\max}(\Theta_\theta(\mathcal{D}, \mathcal{D}))$ and $\lambda_{\max}(\hat{\Theta}_\theta(\mathcal{D}, \mathcal{D}))$ decreases.

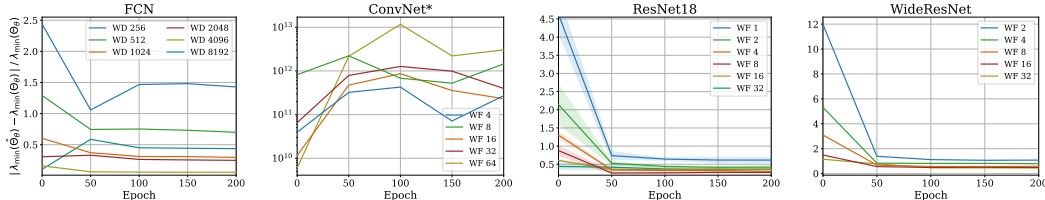

Figure 5: Evaluating the **relative difference of** $\lambda_{\min}$ **of** $\Theta_\theta(\mathcal{D}, \mathcal{D})$ **and** $\hat{\Theta}_\theta(\mathcal{D}, \mathcal{D})$ at initialization and throughout training. The kernels have been evaluated using a subset ($|\mathcal{D}| = 1000$) of points from CIFAR-10. As the NN's width grows, the relative difference in $\lambda_{\min}(\Theta_\theta(\mathcal{D}, \mathcal{D}))$ and $\lambda_{\min}(\hat{\Theta}_\theta(\mathcal{D}, \mathcal{D}))$ decreases. Note the extremely large values reported for ConvNet. As observed previously in Lee et al. (2020) and Xiao et al. (2020), CNN-GAP is ill-conditioned and $\lambda_{\min}(\Theta_\theta(\mathcal{D}, \mathcal{D})) \to 0$, while $\lambda_{\min}(\hat{\Theta}_\theta(\mathcal{D}, \mathcal{D})) > 0.001$, causing the huge discrepancy. More details in Appendix B.3.

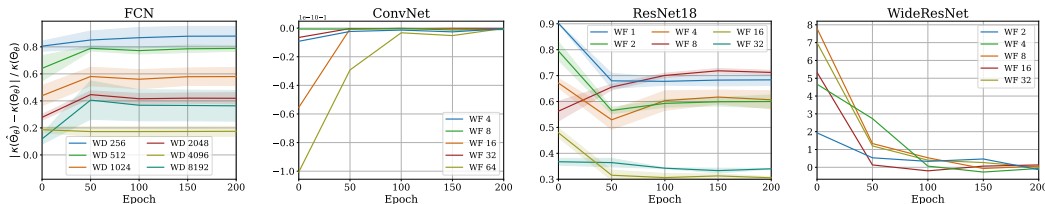

Figure 6: Evaluating the **relative difference of condition number of eNTK and pNTK** at initialization and throughout training. Condition number is defined as $\kappa(K) = \lambda_{\max}(K)/\lambda_{\min}(K)$. The kernels have been evaluated using a subset ($|\mathcal{D}| = 1000$) of points from CIFAR-10. As the NN's width grows, the relative difference in $\lambda_{\min}(\Theta_\theta(\mathcal{D}, \mathcal{D}))$ and $\lambda_{\min}(\hat{\Theta}_\theta(\mathcal{D}, \mathcal{D}))$ decreases. The bizarre values found in the ConvNet plot can be addressed as discussed in Figure 5; more details in Appendix B.3.

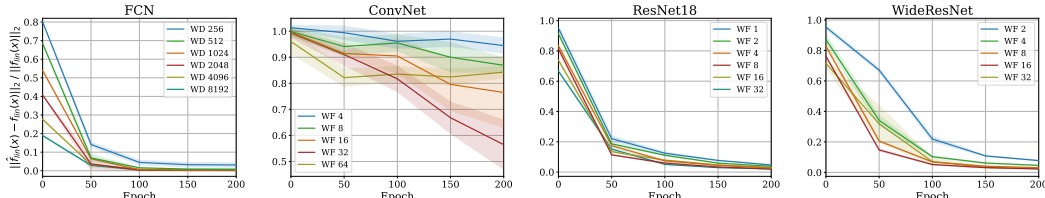

Figure 7: Evaluating the **relative norm difference of kernel regression outputs using eNTK and pNTK as in Equation (5) and Equation (6)** at initialization and throughout training. The kernel regression has been done on $|\mathcal{D}| = 1000$ training points and $|\mathcal{X}| = 500$ test points randomly selected from CIFAR-10's train and test sets. As the NN's width grows, the relative difference in $\hat{f}^{lin}(\mathcal{X})$ and $f^{lin}(\mathcal{X})$ decreases at initialization. Surprisingly, the difference between these two continues to quickly vanish throughout Sthe training process using SGD.

with the experimental results depicted in Figures 2, 4, 6 and 13. In other words, although the kernels $\Theta_\theta$ and $\hat{\Theta}_\theta \otimes I_O$ seem to be diverging in Frobenius norm, eigenspectrum, and so on, kernel regression using those two kernels converges quickly, so that after 50 epochs the difference in predictions almost totally vanishes. We believe further investigation of why this phenomenon is observed could lead to new interesting insights about the training dynamics of NNs.

## 5 KERNEL REGRESSION USING PNTK ON FULL CIFAR-10 DATASET

Thanks to the reduction in time and memory complexity of pNTK over eNTK, motivated by Theorem 5 and experimental findings in Figure 8, we finally evaluate the corresponding pNTKs of the four architectures that we have used in our experimental evaluations in different widths using full CIFAR-

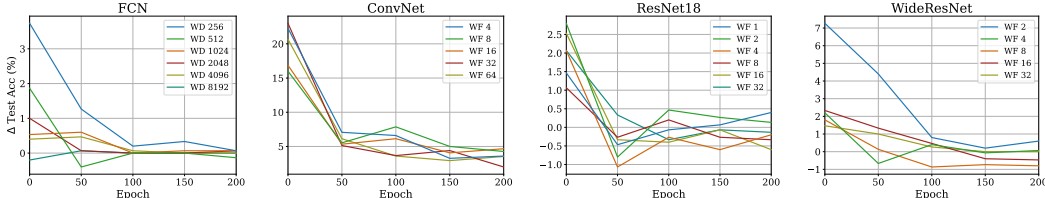

Figure 8: Evaluating the **accuracy difference of kernel regression outputs using eNTK and pNTK as in Equation (5) and Equation (6)** at initialization and throughout training. **Kernel regression using pNTK always achieves a higher test accuracy in comparison to kernel regression using eNTK.** The kernel regression has been done on $|\mathcal{D}| = 1000$ training points and $|\mathcal{X}| = 500$ test points randomly selected from CIFAR-10's train and test sets. As the NN's width grows, the prediction accuracy of in $\hat{f}^{lin}$ and $f^{lin}$ decreases at initialization. Again, the difference between these two continues to vanish throughout the training process using SGD.

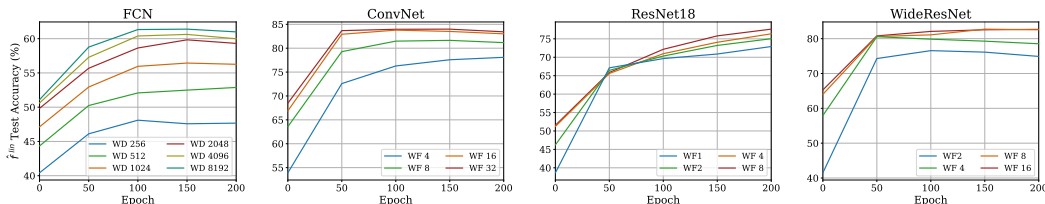

Figure 9: Evaluating the **test accuracy of kernel regression predictions using pNTK as in Equation (6) on the full CIFAR-10 dataset** at initialization and throughout training. As the NN's width grows, the test accuracy of $\hat{f}^{lin}$ is also improved, but eventually saturates with the growing width. Using trained weights in computation of pNTK results in improved test accuracy of $\hat{f}^{lin}$.

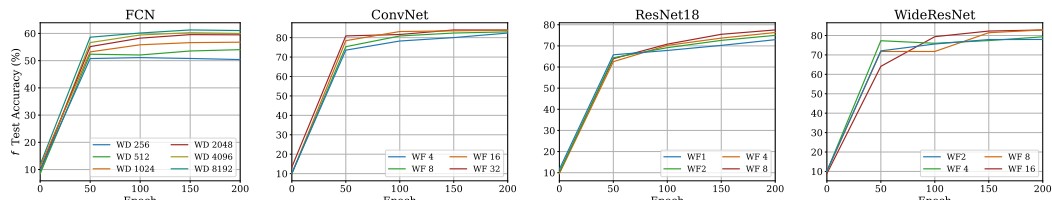

Figure 10: Evaluating the **test accuracy of model $f$ throughout SGD training on the full CIFAR-10 dataset**. In contrast to $\hat{f}^{lin}$, the test accuracy of $f$ does not significantly improve with growing width.

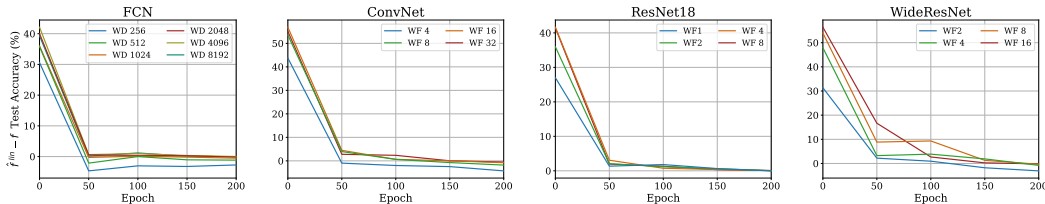

Figure 11: Evaluating the **difference in test accuracy of kernel regression using pNTK as in (6) vs the current model $f$** throughout SGD training on the full CIFAR-10 dataset: how much does a linearized predictor with the current representation improve prediction accuracy over SGD?

10 data, at initialization and throughout training the models under SGD. As mentioned previously, running kernel regression with eNTK on all of CIFAR-10 would require evaluating $25 \times 10^{10}$ Jacobian-vector products and more than $\approx 1.8$ terabytes of memory; using pNTK, this can be done with a far more reasonable $25 \times 10^8$ Jacobian-vector products and $\approx 18$ gigabytes of memory. This is

still a heavy load compared to, say, direct SGD training, but is within the reach of standard compute nodes.

Figure 9 shows the test accuracy of $\hat{f}^{lin}$ on the full train and test sets of CIFAR-10. In the infinite-width limit, the test accuracy of $\hat{f}^{lin}$ at initialization (and later, because the kernel stays constant in this regime) should match the final test accuracy of $f$: that is, the epoch 0 points in Figure 9 would agree with roughly the epoch 200 points in Figure 10. This comparison is plotted directly in Appendix C. Furthermore, the test accuracies of predictions of kernel regression using the pNTK are lower than those achieved by the NTKs of infinite-width counterparts for fully-connected and fully-convolution networks. This is consistent with results on eNTK by Arora et al. (2019a); Lee et al. (2020), although Arora et al. (2019a) studied only a "CIFAR-2" version.

It is worth noting from Figures 9 and 11 that, in contrast to the findings of Fort et al. (2020), we observe that the corresponding pNTK of the NN $f$ continues to change even after epoch 50 of SGD training. Although for fully-connected networks and some versions of ResNet18 this change is not significant, in fully-convolutional networks and WideResNets the pNTK continues to exhibit changes until epoch 150, where the training error has vanished. We remark that Fort et al. (2020) analyzed eNTKs based on only 500 random samples from CIFAR-10, while the pNTK approximation has enabled us to run our analysis on the 100-times larger full dataset.

Lastly, to help the community better analyze the properties of NNs and their training dynamics and avoid wasting computation by redoing this work, we plan to share computed pNTKs for all the mentioned architectures and widths, as well as pNTKs of ResNets with 34, 50, 101 and 152 layers on CIFAR-10 and CIFAR-100 (Xiao et al., 2017) datasets, both at initialization and using pretrained ImageNet (Deng et al., 2009) weights. We hope that our contribution will enable further analyses and breakthroughs towards a stronger theoretical understanding of the training dynamics of deep neural networks.

# 6 DISCUSSION

Our pseudo-NTK approach to approximating the empirical Neural Tangent Kernel has provable bounds, good empirical performance, and multiple orders of magnitude improvement in runtime speed and memory requirements over the direct empirical NTK. We evaluate our claims and the quality of the approximation under diverse settings, giving new insights into the behaviour of the empirical NTK with trained representations. We help justify the correctness of recent approximation schemes, and hope that our rigorous results and thorough experimentation will help researchers develop a deeper understanding of the training dynamics of finite networks, and develop new practical applications of the NTK theory.

One major remaining question is to theoretically analyze what happens to the pNTK or eNTK during SGD training of the network. In particular, the fast convergence of $\hat{f}^{lin}$ and $f^{lin}$ when training the network, as seen in Figures 7 and 8, runs counter to our expectations based on the approximation worsening in Frobenius norm (Figure 7), maximum eigenvalue (Figure 4), and condition number (Figure 6). This seems likely to be very important to practical use of the pNTK.

Perhaps relatedly, it is also unclear why pNTK consistently results in higher prediction accuracies than when kernel regression is done using eNTK, given that our motivation for pNTK is entirely in terms of approximating the eNTK (Figure 8). Intuitively, this may be related to a regularization-type effect: the pNTK corresponds to a particularly limited choice of a "separable" operator-valued kernel (e.g. Álvarez et al., 2012). Separable kernels are a common choice in that literature for both computational and regularization reasons; by enforcing this particularly simple form, we remove many degrees of freedom relating to the interaction between "tasks" (different classes) that may be unnecessary or hard to estimate accurately with the eNTK. This might, in some sense, correspond to a one-vs-one rather than one-vs-rest framework in the intuitive sense discussed in Section 3. Understanding this question in more detail might require a more detailed understanding of the structure of the eNTK at finite width, and/or a much more detailed understanding of the interaction between classes in the dataset with learning in the NTK regime.

Finally, even the pNTK is still rather expensive compared to running SGD on neural networks. It might make for a better starting point than the full eNTK for other speedup methods, however, like kernel approximation schemes or advanced linear algebra solvers (e.g. Rudi et al., 2017).

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

## A    Details of experimental setup

In this section, we present the details on the experimental setup used for the plots depicted in the main body of the paper. As mentioned, the exact width for FCNs have been reported. For WideResNet-16-k we use two block layers, and the initial convolution in the network has a width of $16WF$ where WF is the reported WF. For instance, $WF = 16$ means that the first block layer has a width of 256 and the second block layer has a width of 512. For ResNet18, we also used the same approach, multiplying WF by 16. Thus, when $WF = 4$, the constructed network will have the exact architecture as the classical ResNet18 architecture reported. A WF of 16 means a ResNet18 with each layer being 4 times wider than the original width.

When training the neural networks using SGD, a constant batch size of 128 was used across all different networks and different dataset sizes used for training. The learning rate for all networks was also fixed to 0.1. However, not all networks were trainable with this fixed learning rate, as the gradients would sometimes blow up and give `NaN` training loss, typically for the largest width of each mentioned architecture. In those cases, we decreased the learning rate to 0.01 to train the networks.

Note that to be consistent with the literature on NTKs, techniques like data augmentation have been turned off, but a weight decay of 0.0001 along with a momentum of 0.9 for SGD is used. Data augmentation here plays an important role in the attained test accuracies of the fully trained networks.

## B    Further results on the approximation quality

In this section we'll lay out the proofs of the theorems provided in the main text, mainly Theorems 1, 4 and 5. Towards this, we first define some notation and show a simple recursive formula for computing the tangent kernel that we take advantage of to prove the theorems. Consider a NN $f : \mathbb{R}^D \to \mathbb{R}^O$. We assume the final read-out layer of the NN $f$ is a dense layer with width $w$. Assuming the NN $f$ has $L$ layers, we define $\theta_l$ to be the corresponding parameters of layer $l \in \{1, 2, \ldots, L\}$. Furthermore, let's define $g : \mathbb{R}^D \to \mathbb{R}^w$ as the output of the immediate last layer of the NN $f$, such that $f(x) = \theta_L g(x)$ for some $\theta_L \in \mathbb{R}^{O \times w}$.

As shown by Lee et al. (2019); Yang (2020), the NTK can be reformulated as the layer-wise sum of gradients (when the parameters of each layer $\theta_l$ are assumed to be vectorized) of the output with respect to $\theta_l$. Accordingly, we denote eNTK of a NN $f$ as

$$\Theta_f(x_1, x_2) = \sum_{l=1}^{L} \nabla_{\theta_l} f(x_1) \nabla_{\theta_l} f(x_2)^\top. \tag{8}$$

Now, noting that as the final layer of $f$ is a dense layer, we can use the chain rule to write $\nabla_{\theta_l} f(x)$ as $\frac{\partial f}{\partial g(x)} \frac{\partial g(x)}{\partial \theta_l}$ where $\frac{\partial f(x)}{\partial g(x)} = \theta_L$. Thus, we can rewrite (8) as

$$\begin{aligned}
\Theta_f(x_1, x_2) &= \sum_{l=1}^{L-1} \theta_L \nabla_{\theta_l} g(x_1) \nabla_{\theta_l} g(x_2)^\top \theta_L^\top + \nabla_{\theta_L} f(x_1) \nabla_{\theta_L} f(x_2)^\top \\
&= \theta_L \left( \sum_{l=1}^{L} \nabla_{\theta_l} g(x_1) \nabla_{\theta_l} g(x_2)^\top \right) \theta_L^\top + g(x_1)^\top g(x_2) I_O \\
&= \theta_L \, \Theta_g(x_1, x_2) \, \theta_L^\top + g(x_1)^\top g(x_2) I_O.
\end{aligned} \tag{9}$$

Applying Equation (9), we can already see that the pNTK of a network $f$ simply calculates a weighted summation of all elements of eNTK into a scalar, since it can be seen as adding a new final dense layer to the network $f$ with the fixed weight vector $\frac{1}{\sqrt{O}} \mathbf{1}_O$ where $\mathbf{1}_O$ is the $O$-dimensional vector consisting of all 1s. Note that, however, this $\mathbf{1}_O$ vector is not trainable in this context and is a fixed vector.

Before moving on with the approximation proofs, we would like to mention that the proofs in this section rely heavily on concentration inequalities of *sub-exponential* random variables. Thus, we start by providing some background about sub-exponential random variables and the related concentration inequalities that we will use later on.

### B.1 BACKGROUND ON SUB-EXPONENTIAL RANDOM VARIABLES

A real-valued random variable $X$ with mean $\mu$ is called *sub-exponential* (Wainwright, 2019) if there are non-negative parameters $(\nu, \alpha)$ such that

$$\mathbb{E}[e^{\lambda(X-\mu)}] \leq e^{\frac{\nu^2\lambda^2}{2}} \quad \text{for all } |\lambda| < \frac{1}{\alpha}.$$

We use $X \sim SE(\nu, \alpha)$ to denote that $X$ is a sub-exponential random variable with parameters $(\nu, \alpha)$, but note that this is not a particular distribution.

A famous sub-exponential random variable is the product of two standard normal distributions, $z_i \sim \mathcal{N}(0,1)$, such that the two factors are independent ($X_1 = |z_1||z_2| \sim SE(\nu_p, \alpha_p)$ with mean $2/\pi$) or the same ($X_2 = z^2 \sim SE(2,4)$ with mean 1). We now present a few lemmas regarding sub-exponential random variables that will come in handy in the later subsections of the appendix.

**Lemma 6.** *If a random variable $X$ is sub-exponential with parameters $(\nu, \alpha)$, then the random variable $sX$ where $s \in \mathbb{R}^+$ is also sub-exponential with parameters $(s\nu, s\alpha)$.*

*Proof.* Consider $X \sim SE(\nu, \alpha)$ and $X' = sX$ with $\mathbb{E}[X'] = s\,\mathbb{E}[X]$, then according to the definition of a sub-exponential random variable

$$\mathbb{E}\left[\exp\left(\lambda(X-\mu)\right)\right] \leq \exp(\frac{\nu^2\lambda^2}{2}) \quad \text{for all } |\lambda| < \frac{1}{\alpha}$$

$$\Longrightarrow \mathbb{E}\left[\exp\left(\frac{\lambda}{s}(sX - s\mu)\right)\right] \leq \exp(\frac{\nu^2 s^2 \frac{\lambda^2}{s^2} 2}{2}) \quad \text{for all } |\frac{\lambda}{s}| < \frac{1}{s\alpha} \tag{10}$$

$$\xrightarrow{\lambda' = \frac{\lambda}{s}} \mathbb{E}\left[\exp\left(\lambda'(X'-\mu')\right)\right] \leq \exp(\frac{\nu^2 s^2 \lambda'^2}{2}) \quad \text{for all } |\lambda'| < \frac{1}{s\alpha}$$

Defining $\alpha' = s\alpha$ and $\nu' = s\nu$ we recover that $X' \sim SE(s\nu, s\alpha)$. $\qquad\square$

**Proposition 7.** *If the random variables $X_i$ for $i \in [1-N]$ for $N \in \mathbb{N}^+$ are all sub-exponential with parameters $(\nu_i, \alpha_i)$ and independent, then $\sum_{i=1}^N X_i \in SE(\sqrt{\sum_{i=1}^N \nu_i^2}, \max_i \alpha_i)$, and $\frac{1}{N}\sum_{i=1}^N X_i \sim SE\left(\frac{1}{\sqrt{N}}\sqrt{\frac{1}{N}\sum_{i=1}^N \nu_i^2}, \frac{1}{N}\max_i \alpha_i\right).$*

*Proof.* This is a simplification of the discussion prior to equation 2.18 in Wainwright (2019). $\qquad\square$

**Proposition 8.** *For a random variable $X \sim SE(\nu, \alpha)$, the following concentration inequality holds:*

$$\Pr\left(|X - \mu| \geq t\right) \leq 2\exp\left(-\min\left(\frac{t^2}{2\nu^2}, \frac{t}{2\alpha}\right)\right).$$

*Proof.* The proof directly follows from applying a scalar multiplication to the result derived in Equation 2.18 in Wainwright (2019). $\qquad\square$

**Corollary 9.** *For a random variable $X \sim SE(\nu, \alpha)$, the following inequality holds with probability at least $1 - \delta$:*

$$|X - \mu| < \max\left(\nu\sqrt{2\log\frac{2}{\delta}}, 2\alpha\log\frac{2}{\delta}\right).$$

### B.2 PSEUDO-NTK RELATIVELY CONVERGES TO ENTK AS WIDTH GROWS

Let's denote a neural network with $L$ dense hidden layers whose width is $n$ as:

$$\begin{aligned} f^0(x) &= x \\ f^{l+1}(x) &= \phi(W^{(l+1)}f^l(x)) \\ f(x) = f^L(x) &= W^{(L)}f^{L-1}(x) \end{aligned} \tag{11}$$

such that $\phi$ is a differentiable coordinate-wise activation function.

**Setting A** (ReLU-MLP). We assume the following assumptions hold in our setting:

- We assume $W^{(l)} \in \mathbb{R}^{n_l \times n_{l-1}}$ for $l \in 1, \dots, L$ is initialized according to the He et al. (2015) initialization, meaning that each scalar parameter is distributed according to $\mathcal{N}(0, 1/n_{l-1})$.

- We assume the width of all hidden layers are identical (and equal to $n$). The proof extends naturally to the case of non-equal widths as long as $n_{l+1}/n_l \to c_l \in (0, \infty)$ for each consecutive pair of layers.

- We assume $\phi$ is the ReLU activation. This can be generalized to 1-Lipschitz, ReLU-like functions such as GeLU, PReLU, and so on, as discussed in Appendix B.5.

- We assume the training data $\mathcal{X}$ is finite and contained in a compact set and there are no overlapping datapoints.

**A Note On Parameterization** Although we assume a Gaussian distribution for each scalar variable, the proofs in this section apply to any other distribution used for scalar parameters as long as:

- The variance of the parameters is set according to He et al. (2015), and the mean is zero.
- Each scalar parameter is initialized independently of all other ones.
- The distribution used is a sub-Gaussian.

This applies to all bounded initialization methods, like truncated normal or uniform on an interval. In what follows, we generally assume each $w_{ij}^{l+1} \in SG(1/n_l)$ with mean zero. In general, the product of two sub-Gaussian distributions has a sub-exponential distribution. For the product of two independent weights, $w_{ij}^{l+1} w_{ab}^{l+1}$ with $i \neq a$ and/or $j \neq b$, we denote the parameters as $\frac{1}{n_l} SE(\nu_p, \alpha_p)$. For $(w_{ij}^{l+1})^2$, we use the parameters $\frac{1}{n_l} SE(\nu_s, \alpha_s)$, whose mean is $\mu_s \neq 0$.

Note that we can recursively define the eNTK of $f^{l+1}$ using the eNTK of $f^l$ as

$$
\begin{aligned}
\Theta^{(l+1)}(x_1, x_2) &= \sum_{i=1}^{l} \frac{\partial f^{l+1}(x_1)}{\partial W^{(i)}} \frac{\partial f^{l+1}(x_2)}{\partial W^{(i)}}^{\top} + \overbrace{\frac{\partial f^{l+1}(x_1)}{\partial W^{l+1}} \frac{\partial f^{l+1}(x_2)}{\partial W^{l+1}}^{\top}}^{K_D^{l+1}(x_1, x_2)} \\
&= \sum_{i=1}^{l} \frac{\partial \phi(W^{(l+1)} f^l(x_1))}{\partial W^{(i)}} \frac{\partial \phi(W^{(l+1)} f^l(x_2))}{\partial W^{(i)}}^{\top} + K_D^{l+1}(x_1, x_2) \\
&= \sum_{i=1}^{l} \frac{\partial \phi(W^{(l+1)} f^l(x_1))}{\partial f^l(x_1)} \frac{\partial f^l(x_1)}{\partial W^{(i)}} \frac{\partial f^l(x_2)}{\partial W^{(i)}}^{\top} \frac{\partial \phi(W^{(l+1)} f^l(x_2))}{\partial f^l(x_2)}^{\top} + K_D^{l+1}(x_1, x_2) \\
&= \frac{\partial \phi(W^{(l+1)} f^l(x_1))}{\partial f^l(x_1)} \left[ \sum_{i=1}^{l} \frac{\partial f^l(x_1)}{\partial W^{(i)}} \frac{\partial f^l(x_2)}{\partial W^{(i)}}^{\top} \right] \frac{\partial \phi(W^{(l+1)} f^l(x_2))}{\partial f^l(x_2)}^{\top} + K_D^{l+1}(x_1, x_2) \\
&= \frac{\partial \phi(W^{(l+1)} f^l(x_1))}{\partial f^l(x_1)} \Theta^{(l)}(x_1, x_2) \frac{\partial \phi(W^{(l+1)} f^l(x_2))}{\partial f^l(x_2)}^{\top} + K_D^{l+1}(x_1, x_2)
\end{aligned}
\tag{12}
$$

where

$$
\frac{\partial \phi(W^{(l+1)} f^l(x))}{\partial f^l(x)} = W^{(l+1)} \odot \left[ \dot{\phi}(W^{(l+1)} f^l(x)) \right]_{1 \times n}
\tag{13}
$$

and $K_D^{l+1}(x_1, x_2) = f^l(x_1)^{\top} f^l(x_2) I_n$ is a diagonal matrix. We can think of the last layer as following the same equations with $\phi$ the identity function, so that $\phi'(x) = 1$. Furthermore, note that using the same approach we can show that pNTK of the layer $l$ can be derived as

$$
\hat{\Theta}^{(l+1)}(x_1, x_2) = \frac{\mathbf{1}_n}{\sqrt{n}} \Theta^{(l+1)}(x_1, x_2) \frac{\mathbf{1}_n^{\top}}{\sqrt{n}}
\tag{14}
$$

where $\mathbf{1}_n$ is the vector of 1s with size $n$.

We now sketch the proof idea first and then move onto rigorously proving each part of the sketch. First, note that using Equation (12) we can recursively calculate the eNTK of a general MLP. We take advantage of this recursive definition and derive bounds for the magnitude of the elements of the eNTK on a layer-to-layer basis recursively. To do so, we first show that the eNTK of the first layer of the NN, $\Theta^{(1)}(x_1, x_2)$, is in general a diagonal matrix. Then, we present a series of Lemmas that bound the elements of the eNTK of layer $l + 1$ based on the magnitude (bounds) of the eNTK of layer $l$. Finally, based on the derived bounds on the magnitude of elements of the eNTK of a NN with $l$ layers and Equation (14), we prove that the Frobenius norm of the pNTK relatively converges to the Frobenius norm of the corresponding eNTK with high probability over random initialization.

Before moving on, it's useful to first show a simple inequality on the elements of a tangent kernel based on the Lipschitz-ness of the activation function; this will help us further in deriving the aforementioned bounds. Define $V^{(l)}(x) = W^{(l)} \odot \left[ \dot{\phi}(W^{(l)} f^{l-1}(x)) \right]_{1 \times n}$. We can write each entry of $\Theta^{(l+1)}(x_1, x_2)$ as

$$\Theta^{(l+1)}(x_1, x_2)_{ij} = \sum_{a=1}^{n} \sum_{b=1}^{n} V^{(l+1)}(x_1)_{ia} V^{(l+1)}(x_2)_{jb} \Theta^{(l)}(x_1, x_2)_{ab} + f^l(x_1)^{\top} f^l(x_2) \mathcal{I}(i = j)$$

$$|\Theta^{(l+1)}(x_1, x_2)_{ij}| \leq |\sum_{a=1}^{n} \sum_{b=1}^{n} W^{(l)}{}_{ia} W^{(l)}{}_{jb} \Theta^{(l)}(x_1, x_2)_{ab}| + |f^l(x_1)^{\top} f^l(x_2)| \mathcal{I}(i = j)$$

$$\leq |\sum_{a=1}^{n} \sum_{b=1}^{n} SE_{iajb}(\nu_p, \alpha_p) \Theta^{(l)}(x_1, x_2)_{ab}| + |f^l(x_1)^{\top} f^l(x_2)| \mathcal{I}(i = j) \quad (15)$$

where $\mathcal{I}$ denotes the 0-1 indicator function, the first inequality follows from the activation function $\phi$ being 1-Lipschitz and the second inequality follows from the product of two sub-gaussian distributions being distributed as a sub-exponential variable. Note that we use $SE_{iajb}(\nu_p, \alpha_p)$ for the sake of generality but in the case $i = j \wedge a = b$ it actually refers to $SE_{ia}(\nu_s, \alpha_s)$.

**Lemma 10** (Diagonality of the first layer's tangent kernel). *For a NN under Setting A, the corresponding eNTK of the first layer $\Theta^{(1)}(x_1, x_2)$ is diagonal. Moreover, there is a corresponding constant $C^{(1)} > 0$ such that for all diagonal elements $\Theta^{(1)}(x_1, x_2)_{ii}$, we have that*

$$|\Theta^{(1)}(x_1, x_2)_{ii}| \leq C^{(1)}.$$

*Proof.* Consider the one layer NN $f^1(x) = \phi(W^{(1)} x)$. For this case, we have:

$$\Theta^{(1)}(x_1, x_2)_{ij} = \begin{cases} \sum_{a=1}^{D} x_{1a} \dot{\phi}(W_i x_1) x_{2a} \dot{\phi}(W_i x_2) & \text{if } i = j \\ 0 & \text{if } i \neq j \end{cases} \quad (16)$$

and thus, since the activation function $\phi$ is 1-Lipschitz we can conclude that for all $i, j$

$$|\Theta^{(1)}(x_1, x_2)_{ij}| \leq \begin{cases} |x_1^{\top} x_2| & \text{if } i = j \\ 0 & \text{if } i \neq j \end{cases}. \quad (17)$$

Thus, the tangent kernel of the first layer is a diagonal matrix whose entries are independent of the width of the first layer ($n$), and can be bounded by a positive constant, given by $C^{(1)} = \max_{(x_1, x_2) \in \mathcal{X} \times \mathcal{X}} |x_1^{\top} x_2|$. $\square$

Next, we present a series of lemmas that will help us derive the bounds on the elements of the tangent kernel of layer $l + 1$ based on the bounds of the tangent kernel of layer $l$. The next lemma specifically bounds the values of the tangent kernel of the *second* layer, based on the diagonality of the first layer's tangent kernel and (15).

**Lemma 11.** *Consider a NN under Setting A with depth $\geq l + 1$. Assume there is a constant $C^{(l)} > 0$ such that $|\Theta^{(l)}(x_1, x_2)_{ii}| < C^{(l)}$ for all $i \in [n_l]$ and every non-diagonal element of $\Theta^{(l)}(x_1, x_2)$ is zero. Then for any small $\delta > 0$ there are corresponding constants $C_1^{(l+1)} = \text{polylog}(n/\delta), C_2^{(l+1)} =$*

$\mathrm{polylog}(n/\delta), n^{l+1} > 0$ *such that for any* $n > n^{l+1}$, *it holds with probability at least* $1 - \delta$ *that, simultaneously for all* $i, j$,

$$|\Theta^{(l+1)}(x_1, x_2)_{ij}| \leq \begin{cases} C_1^{(l+1)} n & \text{if } i = j \\ C_2^{(l+1)}/\sqrt{n} & \text{if } i \neq j. \end{cases} \tag{18}$$

*Proof.* Recall that

$$\Theta^{(l+1)}(x_1, x_2)_{ij} = \sum_{a=1}^{n} \sum_{b=1}^{n} V^{(l+1)}(x_1)_{ia} V^{(l+1)}(x_2)_{jb} \Theta^{(l)}(x_1, x_2)_{ab} + f^l(x_1)^\top f^l(x_2) \mathcal{I}(i = j),$$

where

$$V^{(l+1)}(x_1)_{ia} = W^{(l+1)}{}_{ia} \dot{\phi} \left( \sum_{c=1}^{n} W_{ic}^{(l+1)} f^l(x)_c \right).$$

The weight $W^{(l+1)}{}_{ia}$ is sub-Gaussian, and the $\dot{\phi}$ term has absolute value at most one, so their product is also sub-Gaussian with the same variance proxy parameter (as can be easily seen e.g. from the moment-based characterization of sub-Gaussianity). As noted before, when the indices match we replace $V^{(l+1)}(x_1)_{ia} V^{(l+1)}(x_2)ia$ by $\frac{1}{n_l} SE_{ia}(\nu_s, \alpha_s)$ (which has mean $\mu_s \neq 0$). In the case when the indices are not the same, the two are independent but **do not necessarily have mean zero**. However, mean of each decays with $\frac{1}{n_l}$ at least. In what follows, we show this for Gaussian initialization (extension to other sub-Gaussian distributions is straightforward). To show that the mean of each $V_{ia}^{(l+1)}(x)$ decays with $\frac{1}{n_l}$, we introduce an event

$$E : \left\{ |W_{ia}^{(l+1)} f^l(x)_a| > |\sum_{c=1, c\neq a}^{n} W_{ic}^{(l+1)} f^l(x)_c| \ \wedge \ W_{ia}^{(l+1)} f^l(x)_a > 0 \ \wedge \ \sum_{c=1, c\neq a}^{n} W_{ic}^{(l+1)} f^l(x)_c < 0 \right\}$$

and compute the conditional mean on E happening or not, as in

$$\mathbb{E}\left[ V^{(l+1)}(x)_{ia} \right] = \mathbb{E}\left[ V^{(l+1)}(x)_{ia} \,\Big|\, E \right] \Pr[E] + \mathbb{E}\left[ V^{(l+1)}(x)_{ia} \,\Big|\, \neg E \right] \Pr[\neg E]$$

where $\mathbb{E}\left[ V^{(l+1)}(x)_{ia}(x) \,\Big|\, \neg E \right] = 0$ and $\Pr[E] < 1$. Here, $E$ corresponds to the event where the weight scalar $W_{ia}^{(l+1)}$ is correlated with the argument in $\dot{\phi}$ and $\dot{\phi}(\cdot) = 1$. Without loss of generality, we can condition $\mathbb{E}\left[ V^{(l+1)}(x)_{ia} \,\Big|\, E \right]$ on $f^l(x)_a > 0$ and $\sum_{c=1, c\neq a}^{n} W_{ic}^{(l+1)} f^l(x)_c < 0$ (as it indeed follows a zero mean normal distribution, hence introducing a factor of 2 which we omit for simplicity) and calculate

$$\begin{aligned} \mathbb{E}\left[ V^{(l+1)}(x)_{ia} \,\Big|\, E \right] &= \mathbb{E}\left[ W_{ia}^{(l+1)} \,\Big|\, W_{ia}^{(l+1)} > -\frac{1}{f^l(x)_a} \sum_{c=1, c\neq a}^{n} W_{ic}^{(l+1)} f^l(x)_c \right] \\ &= \mathop{\mathbb{E}}_{b \sim \mathcal{N}^+(0, \sigma^2)} \left[ W_{ia}^{(l+1)} \,\Big|\, W_{ia}^{(l+1)} > b \right] \\ &= \frac{1}{n_l} \mathop{\mathbb{E}}_{b \sim \mathcal{N}^+(0, \sigma^2)} \left[ \frac{\phi(b)}{1 - \Phi(b)} \right] \end{aligned} \tag{19}$$

where $\sigma^2 = \frac{\|f^l(x)\|^2 - f^l(x)_a^2}{f^l(x)_a^2 n}$, $\phi$ is the standard normal PDF and $\Phi$ is the standard normal CDF. Note that $\frac{\phi(b)}{1-\Phi(b)} < |b| + 1$. Hence, assuming $\sigma^2 = \mathcal{O}(1)$, $\mathbb{E}\left[ V^{(l+1)}(x)_{ia} \,\Big|\, E \right]$ decays with a rate of $\frac{1}{n_l}$, and so does $\mathbb{E}\left[ V^{(l+1)}(x)_{ia} \right]$. Accordingly, for the rest of the proof, we use the term $\frac{\mu_p}{n_l^2}$ to represent the mean of sub-exponential variable $SE_{iajb}(\nu_p, \alpha_p)$ which stems from product of $V^{(l+1)}(x_1)_{ia}$ and $V^{(l+1)}(x_2)_{jb}$ where the indices $ia$ and $jb$ don't match.

Based on Equation (15) we can expand the elements of $\Theta^{(l+1)}(x_1, x_2)$ as

$$|\Theta^{(l+1)}(x_1, x_2)_{ij}| \leq \begin{cases} |\sum_{a=1}^{n} \sum_{b=1}^{n} W^{(l+1)}{}_{ia} W^{(l+1)}{}_{ib} \Theta^{(l)}(x_1, x_2)_{ab}| + |f^l(x_1)^\top f^l(x_2)| & \text{if } i = j \\ |\sum_{a=1}^{n} \sum_{b=1}^{n} W^{(l+1)}{}_{ia} W^{(l+1)}{}_{jb} \Theta^{(l)}(x_1, x_2)_{ab}| & \text{if } i \neq j \end{cases}$$

$$= \begin{cases} \sum_{a=1}^{n} \frac{1}{n} SE_{ia}(\nu_s, \alpha_s) |\Theta^{(l)}(x_1, x_2)_{aa}| + |f^l(x_1)^\top f^l(x_2)| & \text{if } i = j \\ |\sum_{a=1}^{n} \frac{1}{n} SE_{iaja}(\nu_p, \alpha_p) \Theta^{(l)}(x_1, x_2)_{aa}| & \text{if } i \neq j . \end{cases}$$

(20)

We've assumed an upper bound on the diagonal elements of $\Theta^{(l)}(x_1, x_2)$ of $C^{(l)}$. We have then that

$$\sum_{a=1}^{n} \frac{1}{n} SE(\nu_s, \alpha_s) |\Theta^{(l)}(x_1, x_2)_{aa}| \leq \frac{C^{(l)}}{n} \sum_{a=1}^{n} SE_{ia}(\nu_s, \alpha_s) \sim SE\left(\frac{\nu_s C^{(l)}}{\sqrt{n}}, \frac{\alpha_s C^{(l)}}{n}\right),$$

since each item in the last sum is independent of all others. Noting this term has mean $C^{(l)}\mu_s$, we get a high-probability upper bound via Corollary 9. Since there are $n$ independent such terms, one for each $i \in [n]$, we'll want an upper bound that holds over all of them: with probability at least $1 - \delta_1$ we have

$$\sum_{a=1}^{n} SE_{ia}(\nu_s, \alpha_s) |\Theta^{(l)}(x_1, x_2)_{aa}| \leq C^{(l)}\mu_s + \max\left(2C^{(l)}\sqrt{\frac{2}{n}\log\frac{2n}{\delta}}, \frac{8C^{(l)}}{n}\log\frac{2n}{\delta_1}\right)$$

$$\leq C^{(l)}\left[\mu_s + 2\sqrt{\frac{2}{n}\log\frac{2n}{\delta_1}}\right]$$

for $n$ large enough that $2\sqrt{2} > 8\sqrt{\frac{1}{n}\log\frac{2n}{\delta_1}}$, which will be the case for $n = \Omega\left(\text{polylog}\frac{1}{\delta_1}\right)$.

Likewise, Lemma 17 shows a high-probability upper bound on $|f^l(x_1)^\top f^l(x_2)|$ for all $n > n_m$ with probability at least $1 - \delta_g$. Combining the two with a union bound, we have with probability at least $1 - \delta_1 - \delta_g$ that

$$\max_i |\Theta^{(l+1)}(x_1, x_2)_{ii}| \leq C^{(l)}\left(\mu_s + \sqrt{\frac{8}{n}\log\frac{2n}{\delta_1}}\right) + G^{(l)}n.$$

This inequality is dominated by the $G^{(l)}n$ term. Thus we can find an upper bound $(G^{(l)} + \varepsilon)n$, where for $n \geq \max\left(n_m, \Omega(\text{polylog}(1/\delta_1))\right)$, we have that $\varepsilon = \mathcal{O}\left(\frac{1}{n} + \frac{1}{n^{3/2}}\sqrt{\log\frac{2n}{\delta_1}}\right) = \mathcal{O}\left(\sqrt{\log\frac{1}{\delta_1}}\right)$ can be chosen to be independent of $n$.

For the off-diagonal terms, we have

$$|\Theta^{(l+1)}(x_1, x_2)_{ij}| \leq \frac{C^{(l)}}{n}|\sum_{a=1}^{n} SE_{iaja}(\nu_p, \alpha_p)|.$$

Again applying Corollary 9 for each $i, j$ and taking a union bound gives that with probability at least $1 - \delta_2$,

$$\max_{i,j} |\Theta^{(l+1)}(x_1, x_2)_{ij}| \leq C^{(l)}\left(\frac{\mu_p}{n^2} + \sqrt{\frac{2}{n}\nu_p \log\frac{n(n-1)}{\delta_2}}\right)$$

as long as $n$ is large enough that $\nu_p > \alpha_p\sqrt{\frac{2}{n}\log\frac{n(n-1)}{\delta_2}}$, again true as long as $n = \Omega\left(\text{polylog}\frac{1}{\delta_2}\right)$.

Hence, based on this bound and also the bound provided for the diagonal entries, we can achieve the desired result by setting $n > n^{l+1}$ with probability at least $1 - (\delta_1 + \delta_2 + \delta_g)$. $\qquad\square$

**Lemma 12.** *Consider a NN under Setting A with depth $\geq l+1$. Assume there are constants $C_1^{(l)} = \mathrm{polylog}(n/\delta_p), C_2^{(l)} = \mathrm{polylog}(n/\delta_p)$ such that we have $|\Theta^{(l)}(x_1, x_2)_{ii}| < C_1^{(l)}n$ and $|\Theta^{(l)}(x_1, x_2)_{ij}| < C_2^{(l)}\sqrt{n}$ with probability at least $1 - \delta_p$ and any width $n > n_p$ **simultaneously for all $i,j$.** Then for any arbitrary small $\delta > 0$ there are constants $C_1^{(l+1)} = \mathrm{polylog}(n/\delta), C_2^{(l+1)} = \mathrm{polylog}(n/\delta), n^{l+1} > 0$ such that for any $n > n^{l+1}$ and $i,j$*

$$|\Theta^{(l+1)}(x_1, x_2)_{ij}| \leq \begin{cases} C_1^{(l+1)}n & \text{if } i = j \\ C_2^{(l+1)}\sqrt{n} & \text{if } i \neq j \end{cases} \tag{21}$$

*with probability at least $1 - \delta$. In other words, the magnitude of elements of the tangent kernel in the recursive definition will not grow.*

*Proof.* Using the same expansion that we utilized in the proof for the previous Lemma, for the diagonal elements of $\Theta^{(l+1)}(x_1, x_2)$ we have:

$$|\Theta^{(l+1)}(x_1, x_2)_{ii}| \leq |\sum_{a=1}^{n} \sum_{b=1}^{n} W^{(l+1)}{}_{ia} W^{(l+1)}{}_{ib} \Theta^{(l)}(x_1, x_2)_{ab}| + G^{(l)}n$$

$$= \underbrace{\frac{1}{n}\sum_{a=1}^{n} SE_{ia}(\nu_s, \alpha_s)|\Theta^{(l)}(x_1, x_2)_{aa}|}_{\Theta_1^{(l+1)}(x_1,x_2)_{ii}} + \underbrace{\frac{1}{n}|\sum_{a=1}^{n} \sum_{b=1, b\neq a}^{n} SE_{iaib}(\nu_p, \alpha_p)\Theta^{(l)}(x_1, x_2)_{ab}|}_{\Theta_2^{(l+1)}(x_1,x_2)_{ii}} + G^{(l)}n \tag{22}$$

where

$$|\Theta_1^{(l+1)}(x_1, x_2)_{ii}| \leq \frac{C_1^{(l)}n}{n}\sum_{a=1}^{n} SE_{ia}(\nu_s, \alpha_s) \sim C_1^{(l)}\sqrt{n}SE\left(\nu_s, \frac{\alpha_s\sqrt{n}}{n}\right) \tag{23}$$

with probability at least $(1 - \delta_p)$ and

$$|\Theta_2^{(l+1)}(x_1, x_2)_{ii}| \leq \frac{C_2^{(l)}}{n\sqrt{n}}|\sum_{a=1}^{n} \sum_{b=1, b\neq a}^{n} SE_{iaib}(\nu_p, \alpha_p)|$$

$$\sim \frac{C_2^{(l)}}{n\sqrt{n}}|\sum_{a=1}^{n^2} SE(\nu_p, \alpha_p)| = \frac{C_2^{(l)}}{\sqrt{n}}|SE\left(\nu_p, \frac{\alpha_p}{n}\right)| \tag{24}$$

with probability at least $(1 - \delta_p)$. As shown before, both of these terms are dominated by the $G^{(l)}n$ term in the inequality for diagonal elements and thus, similar to the previous Lemma, we can show that there is an $n^{l+1} = \max(n_m, n_p, \mathrm{polylog}(1/\delta_1))$ (where accordingly $n_m$ is the minimum width coming from Lemma 17) such that for all $n > n^{l+1}$; $|\Theta^{(l+1)}(x_1, x_2)_{ii}| < (G^{(l)}+\varepsilon)n$ with probability at least $1 - (\delta_1 + \delta_p + \delta_g)$ where $\delta_1$ comes from bounding the sub-exponential variables above and $\varepsilon$ can be replaced with a positive constant (which can decay with $\mathcal{O}(n^{-\frac{1}{2}})$). Again, just like the previous Lemma, the same high probability bound (albeit with very slight modifications) holds for all diagonal elements.

For the non-diagonal elements of $\Theta^{(l+1)}(x_1, x_2)$ we have:

$$|\Theta^{(l+1)}(x_1, x_2)_{ij}| = |\sum_{a=1}^{n} \sum_{b=1}^{n} W^{(l+1)}{}_{ia} W^{(l+1)}{}_{jb} \Theta^{(l)}(x_1, x_2)_{ab}|$$

$$\leq \underbrace{\frac{1}{n}|\sum_{a=1}^{n} SE_{iaja}(\nu_p, \alpha_p)\Theta^{(l)}(x_1, x_2)_{aa}|}_{\Theta_1^{(l+1)}(x_1,x_2)_{ij}} + \underbrace{\frac{1}{n}|\sum_{a=1}^{n} \sum_{b=1, b\neq a}^{n} SE_{iajb}(\nu_p, \alpha_p)\Theta^{(l)}(x_1, x_2)_{ab}|}_{\Theta_2^{(l+1)}(x_1,x_2)_{ij}} \tag{25}$$

where

$$|\Theta_1^{(l+1)}(x_1, x_2)_{ij}| \le \frac{C_1^{(l)} n}{n} \sum_{a=1}^{n} |SE_{iaja}(\nu_p, \alpha_p)| \sim C_1^{(l)} \sqrt{n} |SE\left(\nu_p, \frac{\alpha_p \sqrt{n}}{n}\right)| \qquad (26)$$

and

$$|\Theta_2^{(l+1)}(x_1, x_2)_{ij}| \le \frac{C_2^{(l)} \sqrt{n}}{n} |\sum_{a=1}^{n} \sum_{b=1, b\ne a}^{n} SE_{iajb}(\nu_p, \alpha_p)|$$
$$\sim \frac{C_2^{(l)} \sqrt{n}}{n} |\sum_{a=1}^{n^2} SE(\nu_p, \alpha_p)| = C_2^{(l)} \sqrt{n} |SE\left(\nu_p, \frac{\alpha_p}{n}\right)| \qquad (27)$$

each with probability at least $1 - \delta_{in}$. Thus, we can claim that according to Corollary 9 that for all $n > n_p$

$$|\Theta^{(l+1)}(x_1, x_2)_{ij}| \le C^{(l)} \left(\frac{\mu_p}{n^2} + \sqrt{n}\sqrt{2\nu_p \log \frac{2}{\delta_2}}\right) \qquad (28)$$

with probability at least $1 - 2\delta_2$ conditioned on the entries of previous layer's tangent kernel being bounded as mention in the Lemma assumption (Note that we can take advantage of this assumption here as we will include the failure case on the diagonal elements and then have a union bound on diagonal and non-diagonal elements). Here $\delta_2$ comes from applying the Bernstein inequality on the two sub-exponential variables. Applying a union bound on the bounds derived for each entry of the tangent kernel we can see that

$$|\Theta^{(l+1)}(x_1, x_2)_{ij}| \le C^{(l)} \sqrt{n} \left(\frac{\mu_p}{n^2 \sqrt{n}} + \sqrt{2\nu_p \left(\log \frac{4}{\delta_2'} + 2\log n\right)}\right) \qquad (29)$$

for all $i \ne j$ with probability at least $1 - \delta_2'$.

Thus, the Lemma's claim holds with probability at least $1 - (\delta_1 + \delta_2' + \delta_p + \delta_g)$ and $n > n^{l+1}$ as desired.

$\square$

An alert reader already can notice that connecting the previous three Lemmas would result in an upper bound for the diagonal and non-diagonal elements of the eNTK of the NN $f$ at initialization.

**Lemma 13.** *Consider a NN $f$ under Setting A. For every arbitrary small $\delta > 0$, there are constants $C_1 = \mathrm{polylog}(n/\delta), C_2 = \mathrm{polylog}(n/\delta), n_0 > 0$ such that with probability at least $1 - \delta$ over the random initialization, it holds simultaneously for all $i, j$ that if $n > n_0$, the corresponding eNTK of $f$ on the arbitrary datapoints $x_1$ and $x_2$ satisfies*

$$|\Theta(x_1, x_2)_{ij}| \le \begin{cases} C_1 n & \text{if } i = j \\ C_2 \sqrt{n} & \text{if } i \ne j \,. \end{cases} \qquad (30)$$

*Proof.* Starting with Lemma 10, we have a bound for the entries of the first layer's tangent kernel. Plugging this bound into Lemma 11 we get a bound on the elements of the second layer. Next, we can recursively apply Lemma 12. Note that, however, this recursion will induce a new factor in our bound that depends on the depth of our NN, and thus, would enforce the minimum width to also depend on depth. Assume we have applied the first three lemmas and we want to consider the next $L - 3$ recursions of application of Lemma 12 for an NN with $L > 3$ layers. As our assumption, for any arbitrary small $\delta > 0$ there are constants $C_1^{(3)} = \mathcal{O}(\mathrm{polylog}(n/\delta)), C_2^{(3)} = \mathcal{O}(\mathrm{polylog}(n/\delta)), n^{(3)} > 0$ such that for any $n > n^{(3)}$ and $i, j$

$$|\Theta^{(3)}(x_1, x_2)_{ij}| \le \begin{cases} C_1^{(3)} n & \text{if } i = j \\ C_2^{(3)} \sqrt{n} & \text{if } i \ne j \end{cases} . \qquad (31)$$

Now, through recursively applying Lemma 12 we can see that for the layer $L > 3$ for any $\delta > 0$ and $n > n^{(L)}$ there are constants $C_1^{(L)} = \mathcal{O}(\text{polylog}(n(L-3)/\delta)), C_2^{(L)} = \mathcal{O}(\text{polylog}(n(L-3)/\delta)), n^{(L)} > 0$ such that

$$|\Theta^{(L)}(x_1, x_2)_{ij}| \leq \begin{cases} C_1^{(L)} n & \text{if } i = j \\ C_2^{(L)} \sqrt{n} & \text{if } i \neq j \end{cases} \tag{32}$$

with probability at least $1 - \delta$. The change in $n^{(L)}$ is also trackable as presented in Lemma 12 through $\max(n_m^l, n^{(3)}, \Omega(\text{polylog}((L-3)/\delta)))$ for each layer $3 \leq l \leq L$.

$\square$

**Lemma 14.** *Consider a NN $f$ under Setting A. For every arbitrary small $\delta > 0$ and the arbitrary datapoints $x_1$ and $x_2$, it holds that*

$$\|\Theta(x_1, x_2) - \hat{\Theta}(x_1, x_2)\|_F \leq \mathcal{O}(\sqrt{n} \log n) \tag{33}$$

*with probability at least $1 - \delta$ over random initialization for any $n > n_0$ where $n_0 = \text{polylog}(L/\delta)$. In other words, the Frobenius norm of the difference between eNTK and pNTK evaluated on two datapoints are bounded by $\mathcal{O}(\sqrt{n})$.*

*Proof.* We note by $D(x_1, x_2) = \Theta^{(L)}(x_1, x_2) - \hat{\Theta}^{(L)}(x_1, x_2) \otimes I_O$. Using the expansion provided in Equation (14) we can write the

$$|D(x_1, x_2)_{ij}| \leq \frac{1}{n} \begin{cases} |\sum_{a=1}^n \sum_{b=1}^n SE_{iaib}(\nu_p, \alpha_p)\Theta^{(L-1)}(x_1, x_2)_{ab} - \frac{1}{O}\sum_{c=1}^O \sum_{d=1}^O \sum_{a=1}^n \sum_{b=1}^n SE_{cadb}(\nu_p, \alpha_p)\Theta^{(L-1)}(x_1, x_2)_{ab}| \\ |\sum_{a=1}^w \sum_{b=1}^w SE_{iajb}(\nu_p, \alpha_p)\Theta^{(L-1)}(x_1, x_2)_{ab}| \end{cases} \tag{34}$$

where first option is for diagonal elements $i = j$ and second is for non-diagonal ones.

Applying Lemma 13 we can assume there are constants $C_1^{(L-1)} = \text{polylog}(n/\delta_p), C_2^{(L-1)} = \text{polylog}(n/\delta_p)$ such that $|\Theta^{(L-1)}(x_1, x_2)_{aa}| < C_1^{(L-1)} n$ and $|\Theta^{(L-1)}(x_1, x_2)_{ab}| < C_2^{(L-1)} \sqrt{n}$ with probability at least $1 - \delta_p$ **simultaneously** for all $a, b$. Thus we can write the diagonal elements of $D(x_1, x_2)$ as

$$|D(x_1, x_2)_{ii}| \leq \underbrace{C_1^{(L-1)} |\sum_{a=1}^n \left( SE_{ia}(\nu_s, \alpha_s) - \frac{1}{O}\sum_{c=1}^O SE_{ca}(\nu_s, \alpha_s) \right)|}_{D_1(x_1, x_2)_{ii}} + \underbrace{\frac{C_1^{(L-1)}}{O} |\sum_{c=1}^O \sum_{d=1, d \neq c}^O SE_{cada}(\nu_p, \alpha_p)|}_{D_2(x_1, x_2)_{ii}}$$

$$+ \underbrace{\frac{C_2^{(L-1)}}{\sqrt{n}} |\sum_{a=1}^n \sum_{b=1, b \neq a}^n SE_{iaib}(\nu_p, \alpha_p)|}_{D_3(x_1, x_2)_{ii}} + \underbrace{\frac{C_2^{(L-1)}}{O\sqrt{n}} |\sum_{a=1}^n \sum_{b=1, b \neq a}^n \sum_{c=1}^O \sum_{d=1}^O SE_{cadb}(\nu_p, \alpha_p)|}_{D_4(x_1, x_2)_{ii}}.$$

We would like to bound each of $D_k(x_1, x_2)_{ii}$ for $k \in \{1, 2, 3, 4\}$ and then find a bound for the diagonal elements using the combination of them. Starting with $D_1(x_1, x_2)_{ii}$:

$$
\begin{aligned}
|D_1(x_1, x_2)_{ii}| &\leq C_1^{(L-1)} \sum_{a=1}^{n} \left( SE_{ia}(\nu_s, \alpha_s) - \frac{1}{O} \sum_{c=1}^{O} SE_{ca}(\nu_s, \alpha_s) \right) \\
&= C_1^{(L-1)} \sum_{a=1}^{n} \left( SE_{ia}(\nu_s, \alpha_s) - \frac{1}{O} SE_{ia}(\nu_s, \alpha_s) - \frac{1}{O} \sum_{c=1,c\neq i}^{O} SE_{ca}(\nu_s, \alpha_s) \right) \\
&\sim C_1^{(L-1)} \sum_{a=1}^{n} \left( SE\left( \frac{\nu_s(O-1)}{O}, \frac{\alpha_s(O-1)}{O} \right) - SE\left( \sqrt{\frac{(O-1)\nu_s^2}{O^2}}, \frac{\alpha_s}{O} \right) \right) \\
&= C_1^{(L-1)} \sum_{a=1}^{n} SE\left( \sqrt{\frac{\nu_s^2(O-1)^2}{O^2} + \frac{\nu_s^2(O-1)}{O^2}}, \max\left( \frac{\alpha_s(O-1)}{O}, \frac{\alpha_s}{O} \right) \right) \\
&= C_1^{(L-1)} \sqrt{n} SE\left( \nu_s \sqrt{1 - \frac{1}{O}}, \frac{\alpha_s}{\sqrt{n}} \left( 1 - \frac{1}{O} \right) \right)
\end{aligned}
\tag{35}
$$

Thus, using Corollary 9, we can claim

$$
|D_1(x_1, x_2)_{ii}| < C_1^{(L-1)} \sqrt{n} \left( \sqrt{\nu_s \left( 1 - \frac{1}{O} \right) \log \frac{8}{\delta}} + \mu_s \sqrt{1 - \frac{1}{O}} \right)
\tag{36}
$$

with probability at least $1 - \delta/4$.

For the other terms, we can simply the analysis through noting that they are all a form of weighted summation of independent sub-exponential random variables of the same distribution. For such a summation with a weight of $a$ and $b$ summation terms, we have

$$
X = a|\sum_{i=1}^{b} z_i z_i'| \sim a|\sum_{i=1}^{b} SE(\nu_p, \alpha_p)| = a|SE\left( \nu_p \sqrt{b}, \alpha_p \right)|.
\tag{37}
$$

Thus, applying Corollary 9, we get that

$$
|X| < 2a \left( \max\left( \nu_p \sqrt{b \log \frac{8}{\delta}}, \alpha_p \log \frac{8}{\delta} \right) + |\mathbb{E}[X]| \right)
\tag{38}
$$

with probability at least $1 - \delta/4$. Accordingly we can claim

$$
|D_2(x_1, x_2)_{ii}| < 2C_1^{(L-1)} \left( \max\left( \nu_p \sqrt{\left( 1 - \frac{1}{O^2} \right) \log \frac{8}{\delta}}, \alpha_p \log \frac{8}{\delta} \right) + \frac{\mu_p}{n^2} \right),
\tag{39}
$$

$$
|D_3(x_1, x_2)_{ii}| < 2C_2^{(L-1)} \sqrt{n} \left( \max\left( \nu_p \sqrt{\left( 1 - \frac{1}{n^2} \right) \log \frac{8}{\delta}}, \alpha_p \log \frac{8}{\delta} \right) + \frac{\mu_p}{n^2 \sqrt{n}} \right),
\tag{40}
$$

$$
|D_4(x_1, x_2)_{ii}| < 2C_2^{(L-1)} \sqrt{n} \left( \max\left( \nu_p \sqrt{\left( 1 - \frac{1}{O^2} - \frac{1}{n^2} + \frac{1}{O^2 n^2} \right) \log \frac{8}{\delta}}, \alpha_p \log \frac{8}{\delta} \right) + \frac{\mu_p}{n^2 \sqrt{n}} \right),
\tag{41}
$$

all independently and with probability at least $1 - \delta/4$. Moreover, one can easily apply the same technique and see that $D(x_1, x_2)_{ij}$ for $i \neq j$ follows a similar bound to the one of Equation (40).

Thus, loosening the off-diagonal terms for simplicity, applying a union bound on the previous three inequalities yields

$$|D(x_1, x_2)_{ij}| < 8 \left( C_1^{(L-1)} + C_2^{(L-1)} \right) \sqrt{n} \left( \max \left( \nu_p \sqrt{\log \frac{8}{\delta}}, \alpha_p \log \frac{8}{\delta} \right) \right) \tag{42}$$

with probability at least $1 - (\delta + \delta_p)$.

Finally, as $\|D(x_1, x_2)\|_F = \sqrt{\sum_{i,j} D(x_1, x_2)_{ij}^2}$, if each entry's absolute value is less than $t > 0$ then the Frobenius norm is less than $tO$. Thus we can combine a bound on each of the $O^2$ entries to see that

$$\Pr \left( \|D(x_1, x_2)\|_F \leq 8O \left( C_1^{(L-1)} + C_2^{(L-1)} + \mu_s \right) \sqrt{n} \max \left( 2\sqrt{\log \frac{8O^2}{\delta}}, 4 \log \frac{8O^2}{\delta} \right) \right) \geq 1 - (\delta + \delta_p) \tag{43}$$

as desired.

$\square$

**Lemma 15.** *Consider a NN $f$ under Setting A. For every arbitrary small $\delta > 0$ and the arbitrary datapoints $x_1$ and $x_2$, it holds that*

$$\|\Theta(x_1, x_2)\|_F \geq \Omega(n \log n) \tag{44}$$

*with probability at least $1 - \delta$ over random initialization. In other words, the Frobenius norm of the eNTK evaluated on two datapoints is lower bounded by $\Omega(n \log n)$.*

*Proof.* Considering that the dot product of post-activations appear in the diagonal elements of the eNTK in conjunction with Lemma 17, this is straightforward. Note that this bound also applies to the maximum eigenvalue of the eNTK matrix since the maximum eigenvalue is bigger (or equal to) than the sum of elements of the matrix divided by the number of columns. $\square$

We are finally ready to present the proof of Theorem 1.

**Theorem 16.** *Consider a NN $f$ under Setting A. For every arbitrary small $\delta > 0$ and the arbitrary datapoints $x_1$ and $x_2$, there exists $n_0$ such that*

$$\frac{\|\Theta^{(L)}(x_1, x_2) - \hat{\Theta}^{(L)}(x_1, x_2) \otimes I_O\|_F}{\|\Theta^{(L)}(x_1, x_2)\|_F} = \mathcal{O}\left( \frac{1}{\sqrt{n}} \right) \tag{45}$$

*with probability at least $1 - \delta$ for $n > n_0$.*

*Proof.* The proof is straightforward from applying Lemma 14 and Lemma 15. $\square$

**Lemma 17.** *Consider a NN under Setting A with $L \geq 2$ and ReLU activation function. The dot product of two post-activations $|f^{(l)}(x_1)^\top f^{(l)}(x_2)|$ grows linearly with the width of the network with high probability over random initialization.*

*Proof.* We begin by showing that the dot product of the post-activations of the first layer of the NN under setting Setting A grow linearly using a simple Hoeffding bound. Next, we apply *Thorem 1* from Arpit & Bengio (2019) to show that the magnitude of this dot product is preserved in the next layers. First, note that as we assume the data lies in a compact set and as the post-activations are all positive, one can easily see that for each $x_1, x_2 \in \mathcal{X}$ and for all $l \in [1 - L]$ we have that:

$$\min_{x \in \mathcal{X}} \|f^{(l)}(x)\|^2 \leq f^{(l)}(x_1)^\top f^{(l)}(x_2) \leq \max_{x \in \mathcal{X}} \|f^{(l)}(x)\|^2. \tag{46}$$

To simplify the proofs in this Lemma, we use this fact and instead work with the norm of the post-activations and we note that the final result on the norms can be accordingly applied to dot products

of post-activations of different inputs. For the first layer, we have that $f^{(1)}(x) = \sigma(W^{(1)}x)$ where $W_{ij}^{(l+1)} \sim \mathcal{N}(0, \frac{1}{n_l})$ and $x \in \mathbb{R}^{n_0}$. Hence, each $f^{(1)}(x)_i$ is i.i.d and distributed as $\mathcal{N}^R(0, \frac{\|x\|^2}{n_0})$ where $\mathcal{N}^R$ is the Rectified Normal Distribution. Hence, using the properties of the Rectified Normal distribution we get that:

$$\mathbb{E}[\|f^{(1)}(x)\|^2] = \frac{n\|x\|^2}{n_0} \tag{47}$$

Next, as the Rectified Normal is a sub-gaussian distribution, we can apply the Hoeffding bound to see that

$$\left|\|f^{(1)}(x)\|^2 - n\mu_1\right| < \sigma\sqrt{2\log\frac{2}{\delta}} \tag{48}$$

with probability at least $1 - \delta$ where $\mu = \frac{\|x\|^2}{n_0}$ and $\sigma$ is the standard deviation of $\|f^{(1)}(x)\|^2$ over random initialization of the weights of the first layer. Hence, combining this and Equation (46) we can come up with constants $G_1^1$ and $G_2^1$ such that for any $\delta > 0, x_1, x_2, G_1^{(1)}n \leq f^{(1)}(x_1)^\top f^{(1)}(x_2) < G_2^{(1)}n$ with probability at least $1 - \delta$. Next, we can adapt *Theorem 1* from Arpit & Bengio (2019) to see that for post activations of layer $l \in [2 - L]$

$$\Pr\left[(1-\varepsilon)^{l-1}\|f^{(1)}(x)\|^2 \leq \|f^{(l)}(x)\|^2 \leq (1+\varepsilon)^{l-1}\|f^{(1)}(x)\|^2\right] \geq 1 - \delta' \tag{49}$$

where $\delta' = \sum_{l'=2}^{l} 2N \exp\left(-n\left(\frac{\varepsilon}{4} + \log\frac{2}{1+\sqrt{1+\varepsilon}}\right)\right)$, $N$ is the size of our dataset and $\varepsilon$ is any positive small constant. Combining this with the result from the first layer's post-activations we can see that

$$(1-\varepsilon_1)^{l-1}(n\mu - \varepsilon_2) \leq \|f^{(l)}(x)\|^2 \leq (1+\varepsilon_1)^{l-1}(n\mu + \varepsilon_2) \tag{50}$$

with probability at least $(1-\delta)(1-\delta')$. Hence, for any $n > n_m, \delta > 0$ and $(x_1, x_2) \in \mathcal{X} \times \mathcal{X}$ one can come up with constants $G_1^{(l)} = \text{polylog}(n/\delta), G_2^{(l)} = \text{polylog}(n/\delta)$ for post activations of layer $l$ such that

$$G_1^{(l)}n \leq f^{(l)}(x_1)^\top f^{(l)}(x_2) < G_2^{(l)}n \tag{51}$$

with probability at least $1 - \delta$, where $n_m$ depends on $l$ and $\delta$. $\qquad\square$

### B.3 PSEUDO-NTK'S MAXIMUM EIGENVALUE CONVERGES TO ENTK'S MAXIMUM EIGENVALUE AS WIDTH GROWS

In this subsection, we present a formal proof for Theorem 4.

*Proof.* Note that, as both pNTK and eNTK are symmetric PSD matrices, their maximum eigenvalues are equal to their spectral norm. Furthermore, the spectral norm of a matrix is upper-bounded by its Frobenius norm. Now, note that according to the triangle inequality, we have

$$\begin{aligned}\|\Theta(x_1, x_2)\| &= \|\hat{\Theta}(x_1, x_2) \otimes I_O + \left(\Theta(x_1, x_2) - \hat{\Theta}(x_1, x_2) \otimes I_O\right)\| \\ &\leq \|\hat{\Theta}(x_1, x_2) \otimes I_O\| + \|\Theta(x_1, x_2) - \hat{\Theta}(x_1, x_2) \otimes I_O\|\end{aligned} \tag{52}$$

Thus

$$\|\Theta(x_1, x_2)\| - \|\hat{\Theta}(x_1, x_2) \otimes I_O\| \leq \|\Theta(x_1, x_2) - \hat{\Theta}(x_1, x_2) \otimes I_O\|. \tag{53}$$

which according to (43) together with the fact that for any matrix $A$, $\lambda_{\max}(A \otimes I) = \lambda_{\max}(A)$ implies that with probability at least $1 - \delta$,

$$\begin{aligned}\left|\lambda_{\max}\left(\Theta(x_1, x_2)\right) - \lambda_{\max}\left(\hat{\Theta}(x_1, x_2)\right)\right| &\leq \\ 8O\left(C_1^{(L-1)} + C_2^{(L-1)}\right)\sqrt{n}\max&\left(\sqrt{\log\frac{8O^2}{\delta}}, \sqrt{2}\log\frac{8O^2}{\delta}\right).\end{aligned} \tag{54}$$

Moreover, as mentioned in the proof of Lemma 15, combining the previous inequality with the fact that $\lambda_{\max}(\Theta(x_1, x_2)) \geq \Omega(n)$ with high probability shows that there exists $\delta'$ and $n_0$ such that

$$\left| \frac{\lambda_{\max}(\Theta(x_1, x_2)) - \lambda_{\max}\left(\hat{\Theta}(x_1, x_2)\right)}{\lambda_{\max}(\Theta(x_1, x_2))} \right| \leq \mathcal{O}(1/\sqrt{n}) \tag{55}$$

with probability $1 - \delta'$ over random initialization for $n > n_0$ as desired.

$\square$

### B.4 KERNEL REGRESSION USING PNTK VS KERNEL REGRESSION USING ENTK

In this subsection we provide a formal proof for Theorem 5.

*Proof.* We start by proving a simpler version of a theorem, and then show a correspondence that expands the result of the simpler proof to the original Theorem. Assuming $|\mathcal{X}| = |\mathcal{Y}| = N$ (training data), we define

$$h(x) = \Theta(x_1, \mathcal{X})\Theta(\mathcal{X}, \mathcal{X})^{-1}\mathcal{Y} \text{ and } \hat{h}(x) = \left(\hat{\Theta}(x_1, \mathcal{X}) \otimes I_O\right)\left(\hat{\Theta}(\mathcal{X}, \mathcal{X}) \otimes I_O\right)^{-1}\mathcal{Y}. \tag{56}$$

Note that as the result of kernel regression (without any regularization) does not change with scaling the kernel with a fixed scalar, we can use a weighted version of the kernels mentioned in the previous equation without loss of generality. Accordingly, we define

$$\alpha = \left(\frac{1}{n}\Theta(\mathcal{X}, \mathcal{X})\right)^{-1}\mathcal{Y} \text{ and } \hat{\alpha} = \left(\frac{1}{n}\hat{\Theta}(\mathcal{X}, \mathcal{X}) \otimes I_O\right)^{-1}\mathcal{Y}. \tag{57}$$

Using the fact that $\hat{M}^{-1} - M^{-1} = -\hat{M}^{-1}(\hat{M} - M)M^{-1}$ and $(A \otimes I)^{-1} = A^{-1} \otimes I$ we can show that

$$\hat{\alpha} - \alpha = -\hat{\Theta}(\mathcal{X}, \mathcal{X})^{-1} \otimes I_O \left(\frac{1}{n}\hat{\Theta}(\mathcal{X}, \mathcal{X}) \otimes I_O - \frac{1}{n}\Theta(\mathcal{X}, \mathcal{X})\right)^{-1}\Theta(x_1, x_2)\mathcal{Y} \tag{58}$$

Assume $\lambda = \min\left(\lambda_{\min}(\Theta(\mathcal{X}, \mathcal{X})), \lambda_{\min}(\hat{\Theta}(\mathcal{X}, \mathcal{X}))\right)$. Then

$$\|\hat{\alpha} - \alpha\| \leq \frac{1}{\lambda^2}\|\frac{1}{n}\hat{\Theta}(\mathcal{X}, \mathcal{X}) \otimes I_O - \frac{1}{n}\Theta(\mathcal{X}, \mathcal{X})\|\|\mathcal{Y}\| \tag{59}$$

Plugging into the formula for kernel regression, we get that

$$\begin{aligned}
\hat{h}(x) - h(x) &= \left(\frac{1}{n}\hat{\Theta}(x, \mathcal{X}) \otimes I_O\right)\hat{\alpha} - \frac{1}{n}\Theta(x, \mathcal{X})\alpha \\
&= \left(\frac{1}{n}\hat{\Theta}(x, \mathcal{X}) \otimes I_O - \frac{1}{n}\Theta(x, \mathcal{X})\right)\hat{\alpha} + \frac{1}{n}\Theta(x, \mathcal{X})(\hat{\alpha} - \alpha)
\end{aligned} \tag{60}$$

Thus

$$\begin{aligned}
\|\hat{h}(x) - h(x)\| &\leq \|\frac{1}{n}\hat{\Theta}(x, \mathcal{X}) \otimes I_O - \frac{1}{n}\Theta_f(x, \mathcal{X})\|\|\hat{\alpha}\| + \|\frac{1}{n}\Theta(x, \mathcal{X})\|\|\hat{\alpha} - \alpha\| \\
&\leq \frac{1}{\lambda}\|\frac{1}{n}\hat{\Theta}(x, \mathcal{X}) \otimes I_O - \frac{1}{n}\Theta(x, \mathcal{X})\|\|\mathcal{Y}\| \\
&\quad + \frac{1}{\lambda^2}\|\frac{1}{n}\Theta(x, \mathcal{X})\|\|\frac{1}{n}\hat{\Theta}(\mathcal{X}, \mathcal{X}) \otimes I_O - \frac{1}{n}\Theta(\mathcal{X}, \mathcal{X})\|\|\mathcal{Y}\|.
\end{aligned} \tag{61}$$

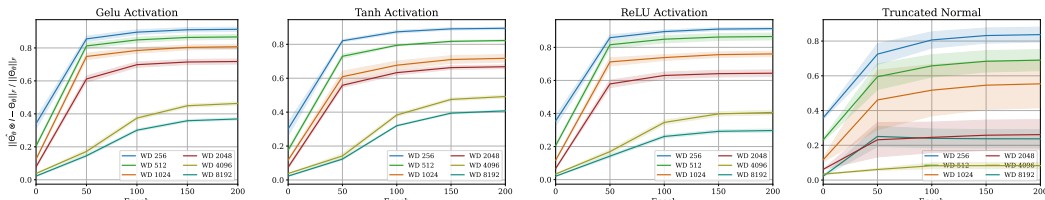

Figure 12: Comparing the **magnitude of sum of on-diagonal and off-diagonal elements of** $\Theta_\theta$ at initialization and throughout training, based on 1000 points from CIFAR-10. The reported numbers are the average of $1000 \times 1000$ kernels each having a shape of $10 \times 10$. The same subset has then been used to train the NN using SGD.

Figure 13: Evaluating the **relative difference of Frobenius norm of** $\Theta_\theta(\mathcal{D}, \mathcal{D})$ **and** $\hat{\Theta}_\theta(\mathcal{D}, \mathcal{D}) \otimes I_O$ at initialization and throughout training, based on 1000 points from CIFAR-10.

Now, note that as for a block matrix $A$ of $A_{ij}$ blocks we have that $\|A\| \leq \sum_{i,j} \|A_{ij}\|$ it follows that for any matrix valued kernel $K$

$$\|K(\mathcal{X}, \mathcal{X})\| \leq \sum_{x_1, x_2 \in \mathcal{X}} \|K(x_1, x_2)\|. \tag{62}$$

Using this fact, we can rewrite the bound as

$$
\begin{aligned}
\|\hat{h}(x) - h(x)\| &\leq \frac{N}{\lambda} \|\frac{1}{n}\hat{\Theta}(x, x_1^*) \otimes I_O - \frac{1}{n}\Theta(x, x_1^*)\|\|\mathcal{Y}\| \\
&\quad + \frac{N^2}{\lambda^2} \|\frac{1}{n}\Theta(x, \mathcal{X})\|\|\frac{1}{n}\hat{\Theta}(x_2^*, x_3^*) \otimes I_O - \frac{1}{n}\Theta(x_2^*, x_3^*)\|\|\mathcal{Y}\|
\end{aligned}
\tag{63}
$$

for some particular $x_1^*, x_2^*, x_3^* \in \mathcal{X}$. Using (43), we can see with probability at least $1 - \delta$ that

$$\left\|\hat{h}(x) - h(x)\right\| \leq \frac{8NO\alpha}{\lambda\sqrt{n}} \max\left(\sqrt{\log\frac{8O^2}{\delta}}, \sqrt{2}\log\frac{8O^2}{\delta}\right) \|\mathcal{Y}\| \left(1 + \frac{N}{\lambda}\|\frac{1}{n}\Theta(x, \mathcal{X})\|\right). \tag{64}$$

To show the correspondence between $\hat{h}(x)$ and $\hat{f}^{lin}(x)$, as in (6), note that

$$
\begin{aligned}
\hat{h}(x) &= \left(\hat{\Theta}(x, \mathcal{X}) \otimes I_O\right)\left(\hat{\Theta}(\mathcal{X}, \mathcal{X})^{-1} \otimes I_O\right)\mathcal{Y} \\
&= \left(\hat{\Theta}(x, \mathcal{X})\hat{\Theta}(\mathcal{X}, \mathcal{X})^{-1} \otimes I_O\right)\mathcal{Y} \\
&= \text{vec}\left(I_O\mathcal{Y}_v\hat{\Theta}(x, \mathcal{X})\hat{\Theta}(\mathcal{X}, \mathcal{X})^{-1}\right)
\end{aligned}
\tag{65}
$$

where $\mathcal{Y}_v = \text{vec}^{-1}(\mathcal{Y})$ is the result of inverse of the vectorization operation, converting the $NO \times 1$ vector to a $O \times N$ matrix. Thus, $\hat{h}(x) = \hat{\Theta}(x, \mathcal{X})\hat{\Theta}(\mathcal{X}, \mathcal{X})^{-1}\mathcal{Y}'$ where $\mathcal{Y}'$ is the $N \times O$ matrix derived from reshaping the $NO \times 1$ vector $\mathcal{Y}$. The proof is complete. □

### B.5 EXTENDING THE PROOFS TO OTHER ARCHITECTURES

In this subsection we elaborate on how one can extend the current proofs to different architectures. We start by providing a sketch on how the dense weight vectors can be replaced by other layers

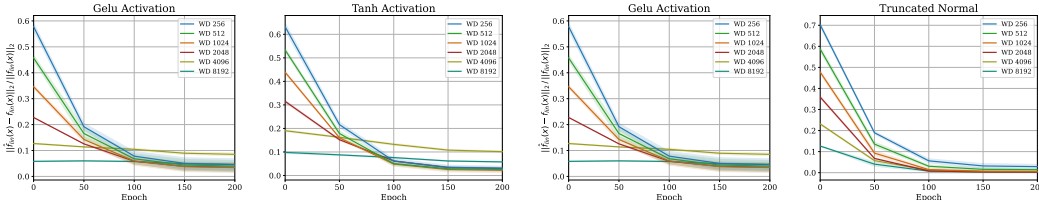

Figure 14: Evaluating the **relative difference of** $\lambda_{\max}$ **of** $\Theta_\theta(\mathcal{D}, \mathcal{D})$ **and** $\hat{\Theta}_\theta(\mathcal{D}, \mathcal{D})$ at initialization and throughout training, based on kernels on a subset ($|\mathcal{D}| = 1000$) of points from CIFAR-10.

Figure 15: Evaluating the **relative norm difference of kernel regression outputs using eNTK and pNTK as in Equation (5) and Equation (6)** at initialization and throughout training. The kernel regression has been done on $|\mathcal{D}| = 1000$ training points and $|\mathcal{X}| = 500$ test points randomly selected from CIFAR-10's train and test sets.

of choice like convolutions. First, note how the linear weights are used in Equation (12). As mentioned in Section 6 of Yang (2020), we can accordingly write the same expansion for other forward computational graphs and derive the corresponding canonincal decomposition for them. In subsection 6.2.1, Yang (2020) provides a concrete example on how one can derive this expansion for a general RNN-like architecture. As the proofs provided in this section depend on the MLP structure only by means of the canonical decomposition, one can extend them to a general architecture by deriving the corresponding canonical decomposition of that architecture.

**Non-Gaussian Weights**: According to the strategy used in the proofs, we need the individual weights to be distributed such that the product of two independent scalar weights (as in Equation (12)) remain sub-exponential. Hence, any sub-gaussian initialization method, such as any bounded initialization (e.g. truncated normal or uniform on an interval) can be used, and the same proof structure would support the same convergence rate, albeit with different constants in convergence (independent of $n$).

**Non-ReLU activations**: In general, the proofs rely on the ReLU activation through Lemma 17, which gives a concentration bound on the absolute value of the dot product of post-activations of each layer of the NN. To use other nonlinearities, we would only need an analagous result for that nonlinearity; the other proofs follow without requiring any other significant change.

**Experimental Evaluation**: To provide further experimental support for this argument, we have conducted an ablation study on the FCN architecture with different nonlinearities and with truncated Gaussian initialization (Figures 12, 13, 14 and 15). As seen in the provided figures, the impact of nonlinearity and initialization method as long as they follow the provided setting in Setting A, is marginal.

## C    MORE DETAILS ON KERNEL REGRESSION USING PNTK ON FULL CIFAR-10 DATASET

In this section we provide another figure comparing the accuracy of $\hat{f}^{lin}(x)$ with parameters derived at epoch $E \in \{0, 50, 100, 150, 200\}$ of training the NN with SGD. On the y-axis, the reported number is $f^{lin}(x) - f^*(x)$ where $f^*$ denotes the final model obtained after training $f$ for 200 epochs. As seen in Figure 16 the architecture of the model has a significant impact on how good the linearization predicts the final accuracy of the fully-trained model. However, as proven in Theorem 1 in conjunction with the linearization approximations provided in Lee et al. (2019), as width grows,

Figure 16: Evaluating the **difference in test accuracy of kernel regression using pNTK as in** (6) **vs the final model** $f$ throughout SGD training on the full CIFAR-10 dataset. How much worse would it be to "give up" on SGD at this point and train $\hat{f}^{lin}$ with the current representation?

this approximation becomes more accurate. One unexplored fact regarding this experiment is that fact that lineraization with trained parameters significantly outperforms linearization at initialization, which is intuitive but not rigorously investigated yet.

