# OpenReview forum: "A Fast, Well-Founded Approximation to the Empirical Neural Tangent Kernel"
_ICLR.cc/2023/Conference — Submitted to ICLR 2023_

### Official Review · Reviewer_JUEa · 2022-10-17

**Confidence:** 3
**Correctness:** 3
**Technical Novelty And Significance:** 2
**Empirical Novelty And Significance:** 3
**Recommendation:** 6

**Clarity, Quality, Novelty And Reproducibility:**

**Clarity**: the paper is in general well written and easy to follow (a few clarifications are needed, see my detailed comments below in "Summary Of The Review."

**Quality and Novelty**: the contributions in the paper are of interest and somewhat novel (theoretical and empirical evaluations of some existing ideas).

**Reproducibility**: The proof is fine. The authors claimed "we plan to share computed pNTKs for all the mentioned architectures ...", which is, however, not present in the current version of the paper.

**Strength And Weaknesses:**

**Strengths**: the paper is in general well written and easy to follow (a few clarifications are needed, though, see my detailed comments below in "Summary Of The Review." The obtained theoretical results are of interest, believed to be true in the literature, but, to the best of my knowledge, rigorously established for the first time. I did not check the proof in great detail, but the line of arguments looks compelling.

**Weaknesses**: the theoretical results obtained in the paper are not very strong and are limited in many aspects (layer width, Gaussian initialization, nonlinearity, etc). Some improvements can be obtained without much effort, see again my detailed comments below.

**Summary Of The Paper:**

In the paper, the author provided some theoretical and experimental results on the so-called "sum of logits" approximation of empirical neural tangent kernel (eNTK) having multiple output units.
The proposed approach consists in approximating the eNTK by a block diagonal matrix as the Kronecker product between some pseudo-NTK approach (pNTK) and $I_O$, the identity matrix of size $O$, if the network has $O$ output units. This saves an order $O^2$ of memory and (up to) order $O^3$ of computation in the evaluation of NTK.
The authors showed, in Theorem 1 and 4, respectively, that the relative difference in Frobenius norm and maximum eigenvalue between the proposed approximation and the true eNTK is of order $O(n^{-1/2})$, for deep networks of width $n$ and with Gaussian random weights. This provides theoretical justifications for the proposed approximation.
Further numerical experiments were provided to validate the proposed approach at and beyond initialization.

**Summary Of The Review:**

I find the contributions of the paper marginally significant or novel. And some clarifications and revisions are needed to help better understand and possibly (re)evaluate the contribution of this work. See below.

Detailed comments:
* P1: I imagine NNGP stands for neural network Gaussian process?
* i'th at the bottom of P3 versus ith in P4
* P4: I get confused when reading "Before turning to the formal results and experimental evaluation, we give some intuition. First, suppose that ...": I suppose that here v_i should be the ith row or column, NOT entry, of a linear read-out layer? Since in the sentence that follows "if the vectors v_i \sim \mathcal N(0, \sigma^2 I_O)"
* Theorem 1: can something be said beyond the current setting? Say, beyond ReLU activation? Different initialization scheme, say, beyond Gaussian? Should we understand \in O(n^{-1/2}) as of order O(n^{-1/2}) as in classical computer science or statistic literature?
* Remark 3 and Figure 2: the referred "ratio of information" here should be explicitly defined and (at least briefly) discussed
* I believe it is necessary to mention in the statement of Theorem 1 that the result holds ONLY for networks having random weights or networks at initialization: in the current version of the theorem this fact is somehow very implicit and this makes a huge difference.
* Theorem 4 and its proof: in fact, the result (and proof) in Theorem 4 can be improved to obtain some control on any corresponding pair of eigenvalues of eNTK and its approximation: it in fact follows from  Weyl's inequality (in linear algebra) that max_{i} |\lambda_i (A) - \lambda_i (B)| \leq \| A - B \|, with \lambda_i(A) eigenvalues of A and \| A \| the spectral/operator norm of A. Some conclusions on the condition number can be drawn in a similar manner.

---

> ### Author Response · Authors · 2022-11-14
> **Response from the authors**
>
> Thanks for your efforts in reviewing our paper.
>
> > The theoretical results obtained in the paper are not very strong and are limited in many aspects (layer width, Gaussian initialization, nonlinearity, etc). Some improvements can be obtained without much effort, see again my detailed comments below.
>
> We understand this concern, and we have added a new subsection (Appendix B.5) that discusses it further. We’ve elaborated on how the same proof holds for most popular initialization methods, and how one can extend the proof to different non-linearities and different architectures like RNNs. Please let us know if this addresses your concerns, and if you have any further comments or points to be addressed.
>
> > The proof is fine. The authors claimed "we plan to share computed pNTKs for all the mentioned architectures ...", which is, however, not present in the current version of the paper.
>
> We indeed have computed these pNTKs and would be willing to share them with the community, but as each of these kernels is a rather large file (i.e. each CIFAR10 kernel is ~20GB), we don’t see how we can possibly share them anonymously. The size limit for the supplementary material is 100MB, and for a Github repo is 5GB. If you know of any convenient means of anonymously sharing these files and would like to do something with them, let us know; otherwise, we’ll share them for the final (non-anonymous) version of the paper.
>
> > P1: I imagine NNGP stands for neural network Gaussian process?
>
> Correct; we’ve clarified.
>
> > P4: I get confused when reading "Before turning to the formal results and experimental evaluation, we give some intuition. First, suppose that ...": I suppose that here v_i should be the ith row or column, NOT entry, of a linear read-out layer? Since in the sentence that follows "if the vectors v_i \sim \mathcal N(0, \sigma^2 I_O)"
>
> Thanks for pointing this out! Yes, $v_i$ is a row of the last linear layer, we’ve clarified.
>
> > Theorem 1: can something be said beyond the current setting? Say, beyond ReLU activation? Different initialization scheme, say, beyond Gaussian?
>
> As mentioned above, Appendix B.5 addresses these concerns accordingly. Please let use know if you have any further concern or if your current concerns are unaddressed.
>
> > Should we understand \in O(n^{-1/2}) as of order O(n^{-1/2}) as in classical computer science or statistic literature?
>
> Yes, we use standard big-O notation.
>
> > Theorem 4 and its proof: in fact, the result (and proof) in Theorem 4 can be improved to obtain some control on any corresponding pair of eigenvalues of eNTK and its approximation: it in fact follows from Weyl's inequality (in linear algebra) that max_{i} |\lambda_i (A) - \lambda_i (B)| \leq | A - B |, with \lambda_i(A) eigenvalues of A and | A | the spectral/operator norm of A. Some conclusions on the condition number can be drawn in a similar manner.
>
> Thank you for the constructive suggestion and detailed analysis. It’s correct that Weyl’s inequality is useful here as we have a bound on the difference of the Frobenius norm. Unfortunately, we think that this doesn’t actually lead to a bound on other eigenvalues of the kernels (pNTK and eNTK) or the condition number in our current setting, since we don’t have any information bounding any individual eigenvalue other than the largest. In other words, we could provide bounds for $\frac{|\lambda_i(pNTK) - \lambda_i(eNTK)|}{|\lambda_j(pNTK) - \lambda_j(eNTK)|}$ for all $i,j$, but with the current information, we don’t see how to bound the condition number. If you see a way to do it, please let us know!

---

> ### Author Response · Authors · 2022-11-17
> **Rebuttal period ends in less than 3 days**
>
> Dear reviewer,
>
> We would like to kindly send you a reminder and ask if you were satisfied with our responses below and our modifications to the manuscript that is recently updated and ask if you have any further concerns or feedbacks, as the discussion window is going to end in less than three days.
>
> Thanks, Authors

---

> ### Author Response · Authors · 2022-11-19
> **Response to reviewer**
>
> Dear reviewer,
>
> As you specifically asked:
>
> > Theorem 1: can something be said beyond the current setting? Say, beyond ReLU activation? Different initialization scheme, say, beyond Gaussian?
>
> We would like to ask you to check our newly uploaded revision, in which we have relaxed the initialization scheme to be any sub-Gaussian initialization method (such as truncated normal, uniform on an interval, etc), and presented experiments on different nonlinearities and initialization methods. Please let us know if this concern is addressed or if it requires further discussion.
>
> Thanks,
> Authors

---

> ### Author Response · Authors · 2022-11-29
> **Looking forward to your response**
>
> Dear Reviewer JUEa,
>
> Could you take a look at our response and let us know if the questions/concerns are well addressed? Any further questions and comments are warmly welcome.
>
>  Thank you!

---

### Official Review · Reviewer_wjkj · 2022-10-24

**Confidence:** 4
**Correctness:** 4
**Technical Novelty And Significance:** 2
**Empirical Novelty And Significance:** 3
**Recommendation:** 5

**Clarity, Quality, Novelty And Reproducibility:**

I feel the paper writing can be further polished: for example, Figure 2 and 3 appear on page 3, but are referred to on page 5. I think the figures can be rearranged.

**Strength And Weaknesses:**

I think this is an interesting work for NTK approximation, and indeed the proposed method can save the memory usage by a factor of $O^2$. However, I am not very sure about novelty, as detailed below.
1. As mentioned on top of page 4 of this paper, some recent work already points out that the standard NTK converges to a diagonal matrix as width goes to infinity. This paper further provides formal theorems (Theorems 1, 4 and 5), but the technical challenge and innovation are not discussed. For example, for convergence in Frobenius norm, I imagine it just follows from randomness of the last layer, and we do not need to deal with some independence issue since other matrices and vectors involved in the gradient calculation do not depend on the last layer, which can simplify the proof a lot. Please correct me if I am wrong, but either way I think it is necessary to discuss the technical difficulties to justify innovation.
2. An interesting result is shown in Figure 7, where the kernel regression outputs of the standard NTK and approximated NTK actually become closer during training. I believe this is a novel discovery, but there is not enough discussion.

**Summary Of The Paper:**

This paper proposes an approximation of the neural tangent kernel (NTK) for models with a large output dimension. Specifically, considering a model with output dimension $O$, for a specific pair of data examples, the standard definition of NTK gives an $O\times O$ matrix. This paper tries to approximate the standard NTK by the identity matrix multiplied with a scalar which is defined as the average of the original NTK. It is proved that as the width goes to infinity, the approximated NTK converges to the standard NTK in Frobenius norm and spectral norm, and the kernel regression solutions are also close. The theoretical results are further supported by experiments.

**Summary Of The Review:**

This paper proposes an efficient way for NTK approximation and provides interesting results, but more justification of novelty is needed.

---

> ### Author Response · Authors · 2022-11-14
> **Response from the authors**
>
> Thanks for your efforts in reviewing our paper.
>
> > As mentioned on top of page 4 of this paper, some recent work already points out that the standard NTK converges to a diagonal matrix as width goes to infinity. This paper further provides formal theorems (Theorems 1, 4 and 5), but the technical challenge and innovation are not discussed. For example, for convergence in Frobenius norm, I imagine it just follows from randomness of the last layer, and we do not need to deal with some independence issue since other matrices and vectors involved in the gradient calculation do not depend on the last layer, which can simplify the proof a lot. Please correct me if I am wrong, but either way I think it is necessary to discuss the technical difficulties to justify innovation.
>
>
>
> In terms of the randomness of the last layer: indeed, this is exactly what we thought when first studying this problem as well. Unfortunately, we were unable to prove a bound based directly on this intuition, but needed to additionally take into account the randomness at each layer: the bound in equation (15), does require the entries of the eNTK of the previous layer to not be too large. Accordingly, we conducted some experiments to see if we can have any assumption for this eNTK in general in different settings (for instance, different architectures, layers at initialization or not, in the case of growing width, etc) and figured out unfortunately the eNTK of the previous layer in general can not be replaced by a constant with high probability. For instance, if the network becomes wider with increasing depth, the entries of the eNTK of the previous layer would also grow with width, and not be bounded by a constant. Hence, we were only able to show results where all layers are at initialization.
>
>
>
> We have revised the proofs in the appendix to explain the different steps taken, the reasoning behind them and the strategies used. We have also elaborated on how the proof can be extended to general architectures, other initializations and non-linearities in the new subsection B.5 in the Appendix, according to reviewers’ suggestions. Please let us know if these address your concerns and if you have any further concern or feedback so that we could revise and improve the paper accordingly.
>
> > An interesting result is shown in Figure 7, where the kernel regression outputs of the standard NTK and approximated NTK actually become closer during training. I believe this is a novel discovery, but there is not enough discussion.
>
>
>
> We totally agree that this finding is a novel discovery and is really interesting. We strongly believe that this discovery deserves more attention, and could be the key to explaining many phenomenons in training neural networks. We have discussed a bit regarding this phenomenon both in Section 4 and also in the Discussion as future work. However, this observation is slightly afar from the goal of this paper, which is the proposal and analysis of pNTK as a fast, well-founded approximation to the eNTK; it seems a bit beyond the scope of this paper to dive deeper into this phenomenon. Moreover, as far as we understand, using the current linearization techniques (like Gronwall-type arguments) to prove bounds between kernel regression on weights that are not necessarily at initialization and the output of neural network when trained starting from those weights would require showing that the minimum eigenvalue of the $\infty$-width NTK on feature learning $\infty$-width networks is positive. Unfortunately, tracking this in general is a very difficult problem.
>
> Please let us know if this addresses your concern, or if you have any further comments or ideas on how to provide a bound for weights that are not at initialization.
>
> > I feel the paper writing can be further polished: for example, Figure 2 and 3 appear on page 3, but are referred to on page 5. I think the figures can be rearranged.
>
> Thanks; we’ve moved the figures in the current revision, and would appreciate any other specific places that you think could be improved with more attention.

---

> ### Author Response · Authors · 2022-11-17
> **Rebuttal period ends in less than 3 days**
>
> Dear reviewer,
>
> We would like to kindly send you a reminder and ask if you were satisfied with our responses below and ask if you have any further concerns or feedbacks, as the discussion window is going to end in less than three days.
>
> Thanks,
> Authors

---

> ### Author Response · Authors · 2022-11-29
> **Looking forward to your response**
>
> Dear Reviewer wjkj,
>
> Could you take a look at our response and let us know if the questions/concerns are well addressed? Any further questions and comments are warmly welcome.
>
> Thank you!

---

> > ### Comment · Reviewer_wjkj · 2022-12-04
> > **response**
> >
> > Dear authors, thanks for your response. If I understand it correctly, one main technical challenge is that we need to show the entries of eNTK are bounded. I agree with that, but I also think it follows from the boundedness of the forward and backward representations, which has also been shown in prior work (e.g., Section 4 of [1]). Can you further elaborate the difference between your techniques and prior techniques?
> >
> > 1. Allen-Zhu, Zeyuan, Yuanzhi Li, and Zhao Song. "A convergence theory for deep learning via over-parameterization." International Conference on Machine Learning. PMLR, 2019.

---

> > > ### Author Response · Authors · 2022-12-05
> > > **Response to Reviewer**
> > >
> > > Dear Reviewer wjkj,
> > >
> > > Thank you for your response.
> > >
> > > Indeed, one of the technical challenges in our work was to derive high probability bounds for the entries of eNTK of a NN. However, it's not the only challenge that we have faced while deriving the final approximation bounds between eNTK and pNTK. In fact, we tried a variety of approaches towards deriving this bound, and in the end realized that it's best to show that entries of eNTK are bounded in an inductive manner (for induction in depth) and then show that this leads to a bound between pNTK and eNTK. This approach takes advantage of the fact that the eNTK of layer $L+1$ can be written as a function of the eNTK of layer $L$.
> > >
> > > **Regarding Allen-Zhu et al. 2019, Section 4:** Theorem 5 analyzes the behaviour of NTK (and kernel regression using it) during training in the setting of ultra-wide NNs in motivation of deriving bounds between the first-order Taylor expansion of a NN and the NN itself while being trained under SGD (hence, difference between eNTK at initialization and shortly after receiving gradient updates). The results of this theorem were also used in [2] to prove similar bounds. This, however, **can not** be used to derive bounds on the eNTK itself. As seen in section 14 (proof of theorem 5), the presented proof takes advantage of the bound presented in Eq. 10.2 which is on the Frobenius norm of the difference of gradients at initialization and shortly after training (in a small ball around initialization, as in the NTK regime the change of parameters is limited).
> > >
> > > Please let us know if this addresses your question, and if you have any other concerns/feedback.
> > >
> > > Sincerely,
> > > Authors
> > >
> > > [2]: On Exact Computation with an Infinitely Wide Neural Net, Sanjeev Arora, Simon S. Du, Wei Hu, Zhiyuan Li, Ruslan Salakhutdinov, Ruosong Wang, 2019.

---

> > > ### Author Response · Authors · 2022-12-08
> > > **Response to Reviewer**
> > >
> > > Dear Reviewer wjkj,
> > >
> > > Did our response clarify the contribution and answer your question? Please also let us know if you have any other concerns/feedback.
> > >
> > > Thanks,
> > > Authors

---

### Official Review · Reviewer_u749 · 2022-10-25

**Confidence:** 3
**Correctness:** 3
**Technical Novelty And Significance:** 2
**Empirical Novelty And Significance:** 2
**Recommendation:** 5

**Clarity, Quality, Novelty And Reproducibility:**

Most parts of the paper are written clearly. However, there are certain inaccurate statements that could be improved, as is mentioned in the Strength And Weaknesses section.

**Strength And Weaknesses:**

Strengths:

Although I did not thoroughly check all the proofs, the proposed methods and the theory look reasonable.

The experiment results are well presented and look convincing.

Weaknesses:

The authors mentioned that the motivation for studying the eNTKs is that it provides a good understanding of a given network’s representation. However, by definition, it seems that whenever one can afford to train a neural network, the cost to calculate the eNTK should also be affordable. Therefore the significance of the proposed method is questionable.

As the authors mentioned, the fact that a eNTK with $NO \times NO$ entries can be approximated by an $N \times N$ kernel matrix is already a well-known result. Therefore the proposed approximation method is not surprising.

Minor comments:

At the top of page 4, the statement "in the infinite width limit the NTK becomes a diagonal matrix" is inaccurate and may be misleading. If I understand correctly, $K(x_i,x_j)$ for $i \neq j$ is not equal to zero.

**Summary Of The Paper:**

This paper proposes a fast approximation of the empirical neural tangent kernel. The authors theoretically prove the approximation accuracy of the proposed method, and use experiments to demonstrate its performance.

**Summary Of The Review:**

The proposed method is reasonable and the theory seems correct. However, the significance and novelty of this paper need further explanation.

---

> ### Author Response · Authors · 2022-11-14
> **Response from the authors**
>
> Thanks for your efforts in reviewing our paper.
>
> > The authors mentioned that the motivation for studying the eNTKs is that it provides a good understanding of a given network’s representation. However, by definition, it seems that whenever one can afford to train a neural network, the cost to calculate the eNTK should also be affordable. Therefore the significance of the proposed method is questionable.
>
> We respectfully disagree with this statement. As noted in [1, 2, 3, 4] and also in https://github.com/google/neural-tangents, it is well known in the community that the computational complexity of computing eNTK of a neural network is orders of magnitude larger than that of training a neural network. In fact, although both of these two involve computing the gradients or Jacobians of the network, backprop only works by summation of gradients for different datapoints and updating parameters accordingly, while eNTK needs to calculate individual Jacobians and then use them in a corresponding outer product. Furthermore, the memory complexity of backprop is roughly on the order of number of parameters ($P$) of the model, while computing the eNTK needs at least $NO * NO$ memory, which is far bigger than $P$ in almost all cases. We believe that Figure 1 in the paper is a good representation of how our proposed approximation is _much_ faster than actual eNTK computation, which itself is faster to compute in most cases in comparison to the $\infty$-width NTK.
>
> > As the authors mentioned, the fact that a eNTK with NO×NO entries can be approximated by an N×N kernel matrix is already a well-known result. Therefore the proposed approximation method is not surprising.
>
> Again, we respectfully disagree with this statement. Although we agree that it is well-known that the corresponding NTK of a network in the $\infty$-width limit is diagonal and hence can be represented as an $N \times N$ kernel, we have elaborated in the introduction on how this is not the case for the finite-width networks. This is further illustrated in Figure 2 in the paper. Hence, based on the provided speedup, the reduced memory complexity, the experimental behaviour and theoretical support, we believe this work is indeed helpful for the community. We have elaborated on this more in the Related Work section and also in the response to reviewer bBN3. Moreover, we are unaware of any work that discusses a similar approximation, and we would be happy to know if there is one, as we believe our work is original and novel in this manner.
>
> > At the top of page 4, the statement "in the infinite width limit the NTK becomes a diagonal matrix" is inaccurate and may be misleading. If I understand correctly, K(xi,xj) for i≠j is not equal to zero.
>
> We agree with your statement. For an $O$-dimensional output network, however, the NTK for any pair of inputs is an $O \times O$ matrix; it is this matrix that becomes diagonal. We've clarified this in the revised paper.
>
> > The proposed method is reasonable and the theory seems correct. However, the significance and novelty of this paper need further explanation.
>
> In addition to this comment, please see our discussion of the significance and novelty in our responses to other reviewers (particularly Reviewer bBN3, response 2/3). We can further emphasize this in revising the paper if you think that will help.
>
> [1]: On Exact Computation with an Infinitely Wide Neural Net, Arora et al. 2019
>
> [2]: Deep learning versus kernel learning: an empirical study of loss landscape geometry and the time evolution of the Neural Tangent Kernel, Fort et al. 2020
>
> [3]: Fast Finite Width Neural Tangent Kernel, Novak et al. 2022
>
> [4]: More Than a Toy: Random Matrix Models Predict How Real-World Neural Representations Generalize, Wei at al. 2022

---

> ### Author Response · Authors · 2022-11-17
> **Rebuttal period ends in less than 3 days**
>
> Dear reviewer,
>
> We would like to kindly send you a reminder and ask if you were satisfied with our responses below and ask if you have any further concerns or feedbacks, as the discussion window is going to end in less than three days. Please let us know so that in case your concerns are not addressed or you have newly raised questions or concerns, so that we have enough time to address them accordingly in the rebuttal window.
>
> Thanks, Authors

---

> ### Author Response · Authors · 2022-11-29
> **Looking forward to your response**
>
> Dear Reviewer u749,
>
> Could you take a look at our response and let us know if the questions/concerns are well addressed? Any further questions and comments are warmly welcome.
>
> Thank you!

---

### Official Review · Reviewer_bBN3 · 2022-10-28

**Confidence:** 3
**Correctness:** 3
**Technical Novelty And Significance:** 3
**Empirical Novelty And Significance:** 2
**Recommendation:** 5

**Clarity, Quality, Novelty And Reproducibility:**

The originality of the work is difficult for me to judge as I am a bit out of practice with this side of the literature, see my question above.

The clarity needs to be improved (it is not a disaster like many submissions, but there does seem to be some polishing to do).

Point 1 from weaknesses (details):

Does \delta_{in} depend on both i and n in Lemma 11?  In the line after equation (21), not only did the authors not define $\delta_g$ (though it is straightforward that it comes from Lemma 18), the delta in the the quantity in the text $(1-\delta)(1-\delta_g)$ cannot be the same as the one showing up in equation (21) itself: in reality there are three sources of failure: the one from lemma 18 ($\delta_g$), the one from the subgaussianity of the first term of equation (20) (let's call this "$\delta_1$") , and the one from the induction hypothesis  (i.e. $\delta_{in}$). The delta in equation (21) should be $\delta_1$, whilst the one in the equation in text should be $\delta_1+\delta_{in}$. There is no way to make this correct even by hiding behind the vagueness of the phrase "where delta depends on delta_in".

Every layer contains failure events, and the $n_l$'s (the minimum values of n which ensure that the results hold) are defined iteratively. **Every $n_l$ also depends on the  previous delta. **
**There is a union bound over the failure events of each layer missing, which should incur at least a factor of $log(L)$ where $L$ is the depth of the network. In addition, the dependence of the minimum width on the final $\delta$ should be quite complicated (though I agree it is still polylog).**  All of this should be worked out more precisely than it has been done here by throwing everything under the carpet term "high" probability. Note that the minimum required width should also depend on the depth.


Point 2:

One of the key calculations in the induction process is arguably inside the proof of Lemma 13, equation (35), where there is a simplification between factors of n, some of which refer to the width of the previous layer whilst some of them refer to that of the next layer. The authors do mention that they are assuming that the widths are within a constant factor of each other, which is indeed necessary for this step to hold. *However, this step needs to hold for the last layer as well, which requires assuming that $O\leq N$, a significant unstated assumption.*



===============Other less key complaints======


There are a few other mid-sized issues with the proofs:

Lemma 12 is not necessary as we can use Lemma 13 instead (the proofs are nearly identical).

In the proof of Lemma 11, one uses Lemma 18, which should probably be put before. The quantity $G^{l}$ is from that Lemma but is not properly introduced. Furthermore, between equation (21) and the end of the ensuing paragraph, $G^{l}$ needs to change! This is relevant because this means that after running through the iterations of applying lemmas 11, 12, 13 etc, we should end up with a quantity $G^{l}$ that is quite different from the $G^{l}$ in Lemma 18.

The proof of Lemma 18 itself is quite problematic. Honestly, I am convinced of the result, but I could only convince myself by looking at the statement and trying to prove it for myself. The proof provided here is very confusing. Certainly the first equality in equation (51) doesn't hold as the variables are not independent. I think this can be fixed with a polarization identity, but that requires a different argument. Note that $\mu$ should also be $\mu^2$ in equation (51). There is also a high probablity statement which is not very thoroughly explained. For instance, when the authors say "with bounded variance", it seems they actually want to refer to the subgaussianity. Just finite variance would not be enough to make the quantities $G^l$ depend logarithmically on $\delta$. Again, the "linear" growth of the quantity bounded in Lemma 18 actually depends on the high probability events considered. One has to mentally assume that all the failure probabilities only end up inside the final estimates in a logarithmic form (which I believe, but is not worked out rigorously).














**Strength And Weaknesses:**

Strengths:
   -This is a very cool topic. I am a little out of touch with the most recent NTK literature but it seems hard to find non-asymptotic results expressed as a function of the width.
   -There is a good mix of theory and practice and assuming everything is correct, this paper can have an impact both in statistical learning theory and in practical applications.


Weaknesses:

  - It is not clearly stated whether or not the main result (Theorem 1, equation (3)) holds only because of the results in this paper or simply because both quantities inside the norm in the numerator converge to the actual (non empirical) NTK at the same rate. I went through the original NTK paper (Jacot et al 18) and it seems like the results are purely asymptotic, but it is still surprising to me that I cannot find a convergence rate (in terms of width) for the eNTK to the NTK. Can the authors provide the relevant references and explain whether their result follows directly from such a non-asymptotic result?

- The proofs are at best not reader friendly and at worse, wrong: there are three main issues as far as I can see (see more details in "clarity, ...")

1. The management of the high probability events (the "deltas") is very vague. I am quite confident that this can be fixed (possibly by also changing the wording of the main theorems slightly), but since the proofs are not extremely impressive in the first place, it doesn't seem out of place to request a picture perfect first submission for a conference such as ICLR. **Honestly, I think the results are (at least slightly) wrong.** There are a few union bounds arguments missing, which should make a factor of log(L) appear.

2. The results assume a constant width in the proofs, quickly saying that using other works' would allow one to make the results architecturally robust. It would be nice to see those techniques in action. This is especially relevant in the sense that I think there is a hidden assumption here: I cannot see how to fix the proof of the main theorem without assuming that $O\leq n$ (the number of outputs is less than the width of the network). This definitely restricts the setting and removes some of the typical Xtreme multiclass cases.

3. The explanations in the proofs are rushed and it can be hard to read some of them. This is despite the fact that the main strategy of the proof is actually pretty straightforward.

**Summary Of The Paper:**

This paper studies the relationship between the "pseudo-NTK" and the "empirical NTK". They are different from each other only in the multi-class (multi-output) case: let $f^L_1,f^L_2, \ldots,f^L_O$ denote the $O$ outputs of the network at the last layer $L$.
If $x_1$ and $x_2$ are two samples,  $eNTK(x_1,x_2)_{i,j}$

is the inner product between the gradients of $f^L_i$ at $x_1$ and that of $f^L_j$ at $x_2$. On the other hand, the $pNTK(x_1,x_2)$ is a single number equal to the inner product between the gradients of $1/\sqrt{O}\sum_{i=1}^O f^L_i$ at $x_1$ and $x_2$.  It is shown here that the pNTK approximated the diagonal elements of the eNTK, and the off diagonal elements of the eNTK are an order of $1/\sqrt{n}$ smaller (than the diagonal elements), where $n$ is the width of the network. This allows one to construct a faster approximation to the eNTK in the multi-output case. It is also shown that the approximation stays good even if one solves the kernel regression problem associated to the relevant kernels, and many data experiments confirm the findings (though obviously not in terms of exact convergence rates. Some of the experiments show that the approximation holds even after training, though this is not proved from a theoretical standpoint.

The proof of the main result works by induction (similarly to other NTK papers), iteratively bounding terms in key equation (15), except that one is not taking expectations but instead leveraging the sub-gaussianity of the relevant variables to bound the activations' absolute values with high probability.

**Summary Of The Review:**

This is an interesting paper in a very hot topic. I cannot 100 percent vouch for originality but if the results are indeed original, they are very worthy of consideration. There are correctness issues. Nothing fatal, but the paper needs polishing.




=======================minor comments, typos=============


Main paper:

page 4: "casting doubt on the correctness of previous results". This type of provocative statement should be used parsimoniously. I know the authors end up saying that the main contribution of this work is to remove the doubt but still, I would advise reformulating that statement.


Page 4 " we prefer the sum-of-logits for our networks..." The use of the word "prefer" makes it difficult to see your main point there. The one versus rest analogy probably needs more explanation as well.

In the experiments (cf. explanation in section 4), the authors seem to say that they provide error bars based on only three samples. Surely this can't be right!

In remark 3, "on diagonal" and "off diagonal" should be swapped.


Appendix:
At the beginning of section B:   $g:\mathbb{R}^O\rightarrow \mathbb{R}^w$ should be $g:\mathbb{R}^D\rightarrow \mathbb{R}^w$

At the bottom of the same page (13), it would be nice to explain that the vector $1_O$ is not trainable (its gradient doesn't show in the kernel).

In the second point of "setting" on page 15, there is the condition that the layers should be within a constant ratio of each other. This is not formulated correctly: mathematically, the condition as formulated there is satisfied for any network as long as the widths are integers.


At the bottom of page 17, there is  $W^{(2)}_{ia}$

which should be


$W^{(l+1)}_{ia}$


(line spacing due to formatting difficulties with tex in openreview)


In the proof of Theorem 17 (which is a reformulation of the main theorem of the paper), the proof consists in two cross references, one of which is "??".

---

> ### Author Response · Authors · 2022-11-14
> **Response from the authors [1/3]**
>
> Thanks for your very thorough reading of our paper; we really appreciate the detailed feedback on our proofs, which will very much improve the final version of the paper. We have incorporated your suggestion into the newly uploaded revision and responded to the questions in this thread. If anything remains unclear or you think it needs further discussion, please do continue the conversation.
>
> > It is not clearly stated whether or not the main result (Theorem 1, equation (3)) holds only because of the results in this paper or simply because both quantities inside the norm in the numerator converge to the actual (non empirical) NTK at the same rate. I went through the original NTK paper (Jacot et al 18) and it seems like the results are purely asymptotic, but it is still surprising to me that I cannot find a convergence rate (in terms of width) for the eNTK to the NTK. Can the authors provide the relevant references and explain whether their result follows directly from such a non-asymptotic result?
>
> Indeed the results of Jacot et al., and most NTK papers, are purely in the infinite width-limit. The only result we’re aware of for finite networks is [6] (cited in the paper as Arora et al. 2019a), who show for _scalar_ output networks (their Theorem 3.1, in our notation with a fixed depth) that $\lvert \Theta(x_1, x_2) - \lim_{n\to\infty} \Theta(x_1, x_2) \rvert = \mathcal O\left( \frac{1}{n^{1/4}} \right)$.
>
> Using the argument you suggested, we would get a bound that’s the sum of two terms: the convergence of the $O$-output eNTK to the NTK, plus $1/\sqrt{O}$ times the convergence of eNTK of the scalar-output eNTK to its NTK. The second term is addressable by the result of [6], getting a rate of $\mathcal O\left( \frac{1}{O n^{1/4}} \right)$. We’re not aware of a paper that’s analyzed the convergence rate of an $O$-dimensional eNTK to its NTK, but if such a result exists, then that could prove an overall bound. This would necessarily get a worse rate than our result, though, since the scalar-output term already has a worse rate.
>
>
> > The proofs are at best not reader friendly and at worse, wrong: there are three main issues as far as I can see (see more details in "clarity, ...")
>
> Thanks again for your very detailed feedback here. We’ve revised our proofs to address the errors you’ve identified, as well as clarifying many related points. To address each concern individually:
>
>
> > The management of the high probability events (the "deltas") is very vague. […] There are a few union bounds arguments missing, which should make a factor of log(L) appear.
>
>
>
> We agree that the management of high probability events was very vague and dropped dependence on some parameters. We’ve revised the handling of these probabilities to track the total failure probabilities appropriately. Please take another look and let us know if we’ve addressed these concerns.
>
> > The results assume a constant width in the proofs, quickly saying that using other works' would allow one to make the results architecturally robust. It would be nice to see those techniques in action.
>
>
>
> We’ve added a new Appendix B.5 about extending the proofs to other architectures using the techniques of Yang (2020); please let us know if this addresses your concerns.
>
> > This is especially relevant in the sense that I think there is a hidden assumption here: I cannot see how to fix the proof of the main theorem without assuming that O≤n (the number of outputs is less than the width of the network). This definitely restricts the setting and removes some of the typical Xtreme multiclass cases.
>
> We believe that the assumption $O \le n$ is not needed; see later discussion.
>
> > The explanations in the proofs are rushed and it can be hard to read some of them.
>
>
>
> We've tried to elaborate more on the flow of the proof in the newly uploaded revision. If you have any suggestions for places that need more explanation, let us know and we’d be happy to add some more.
>
> > The originality of the work is difficult for me to judge
>
> As mentioned above and also in the related works section, we believe this is the first work to theoretically or experimentally evaluate any approximation for the eNTK of finite-width networks. Previous papers have only used such approximations, either with no explicit justification or just using arguments about the approximation being exact in the infinite-width limit.

---

> ### Author Response · Authors · 2022-11-14
> **Response from the authors [2/3]**
>
> > Point 1 [union bound over failure probabilities]
>
> Again, we’re very grateful for your detailed comments on these points. We’ve slightly changed the structure of our Lemmas 10, 11 and 12 in the revised version. First, we don’t actually need any probabilistic statement in Lemma 10 since it’s implied by the compact support of the dataset. Second, we removed the old Lemma 12 as you suggested (the new Lemma 12 is the previous Lemma 13). For Lemmas 11 and 12, we’ve revised the management of high-probability events, and give more detailed analysis on how to set $\delta$ and the minimum required width of the network based on the depth, $\delta$ and the input minimum width. Moreover, we have included the union bound on the depth in Lemma 13, and have elaborated accordingly. Please let us know if these changes address your concerns – we do think they’ve significantly improved the proofs.
>
>
> > One of the key calculations in the induction process is arguably inside the proof of Lemma 13, equation (35), where there is a simplification between factors of n, some of which refer to the width of the previous layer whilst some of them refer to that of the next layer. The authors do mention that they are assuming that the widths are within a constant factor of each other, which is indeed necessary for this step to hold. However, this step needs to hold for the last layer as well, which requires assuming that O≤N, a significant unstated assumption.
>
> The only place that $O$ affects our bounds is in the final result of Lemma 14, where we take a union bound over $O$ terms and hence have a $\log O$ factor in the final bound. As far as we can tell (please tell us if we’ve missed somewhere else), this doesn’t mean that we require $n > O$ (even up to a constant factor) anywhere in the proofs, although indeed (like all NTK analyses) we do not consider the case where $O$ grows with $n$.
>
>
> > Lemma 12 is not necessary as we can use Lemma 13 instead (the proofs are nearly identical).
>
> Thanks; we’ve removed it.
>
> > $G^l$ needs to change!
>
> We’ve significantly clarified the discussion here in the new proof of Lemma 11. We added a description of the result, but think it’s clearer to keep the lemma in its previous place to avoid distracting from the main thrust of the proofs; if you still think it should move earlier we can of course do that. More importantly, it’s indeed true that we need to use a slightly larger value for the constant than that provided by Lemma 17 (previously 18), but we now rigorously track this. Thanks again for pointing this out.
>
>
> > The proof of Lemma 18 itself is quite problematic
>
> Thank you for pointing this out as well. We’ve fixed the proof of the lemma (now 17) based on a result of Arpit and Bengio (2019).
>
>
> > This is an interesting paper in a very hot topic. I cannot 100 percent vouch for originality but if the results are indeed original, they are very worthy of consideration. There are correctness issues. Nothing fatal, but the paper needs polishing.
>
> We are glad to hear that you find our work interesting and very worth of consideration. We have tried to address all the correctness issues according to your suggestions and would be glad to know about any further concerns or feedback to further improve the paper. To reiterate, we believe that this is an original work that is actually very important for the community. We’ve elaborated on the importance of the finite-width eNTK matters in the relevant work, and we are continuing to see a lot of new works since the submission of this work that involve the finite-width eNTK in some context: [1, 2, 3, 4, 5], and many more! We believe the community will see continued use of the finite-width eNTK, and the fact that computing the actual eNTK on multi-class tasks is mostly infeasible, could introduce a huge burden in this regard. Hence, we think our work (or any other work that potentially results in such computational speedup and relaxation of memory requirements and is theoretically supported; we don’t know of any other than ours!) is very beneficial for the community to enable and speed up these kind of analysises.
>
> > page 4: "casting doubt on the correctness of previous results". This type of provocative statement should be used parsimoniously.
>
> We’ve rephrased.
>
> > Page 4 " we prefer the sum-of-logits for our networks..." The use of the word "prefer" makes it difficult to see your main point there.
>
> We’ve rephrased to be clearer about what we know based on our paper, and what’s speculation.

---

> ### Author Response · Authors · 2022-11-14
> **Response from the authors [3/3]**
>
>
> > In the experiments (cf. explanation in section 4), the authors seem to say that they provide error bars based on only three samples. Surely this can't be right!
>
> These experiments are quite computationally expensive; the error bars are indeed based only on three runs. We think this still provides more information than only using one run, which much work in comparable settings in the field does. Note that our error bars are standard errors, and hence incorporate the fact that we only have a small number of runs.
>
>
> > In remark 3, "on diagonal" and "off diagonal" should be swapped.
>
> Thanks; fixed.
>
> **Overall, we thank you for your valuable suggestions, concerns and points that you made. We would be happy to further discuss any possible question or feedback that you might have on our responses to further address possible concerns and enhance the quality of the work.**
>
>
> [1]: A Kernel-Based View of Language Model Fine-Tuning, Malladi et al. 2022
>
> [2]: A Solvable Model of Neural Scaling Laws, Maloney et al. 2022
>
> [3]: TCT: Convexifying Federated Learning using Bootstrapped Neural Tangent Kernels, Yu et al. 2022
>
> [4]: The Eigenlearning Framework: A Conservation Law Perspective on Kernel Regression and Wide Neural Networks, Simon et al. 2021
>
> [5]: Universal characteristics of deep neural network loss surfaces from random matrix theory, Baskerville et al. 2022
>
> [6]: On Exact Computation with an Infinitely Wide Neural Net, Arora et al. 2019
>
> [7]: Deep learning versus kernel learning: an empirical study of loss landscape geometry and the time evolution of the Neural Tangent Kernel, Fort et al. 2020

---

> ### Author Response · Authors · 2022-11-17
> **Rebuttal period ends in less than 3 days**
>
> Dear reviewer,
>
> We would like to kindly send you a reminder and ask if you were satisfied with our responses below and the new version of the proofs available in the Appendix and ask if you have any further concerns or feedbacks, as the discussion window is going to end in less than three days. We would be more than happy to continue the discussion and address any further concerns that you might have.
>
> Thanks,
> Authors

---

> > ### Comment · Reviewer_bBN3 · 2022-11-18
> > **Answer to rebuttal**
> >
> > Dear Authors,
> >
> > Thanks for your very detailed rebuttal. I like how you colored the revisions in blue.  I can also see that you are very knowledgeable about the topic and my opinion of the work improved. I am sure the authors are competent fundamentally and have a good understanding of what is going on in their own work, which is often not the case in many rejected submission. I do feel they need to sit down and work on the details though.
> >
> >
> >  Here are my comments and concerns:
> >
> >
> > 1. Thanks for improving the union bounds  in Lemmas 11 and 12.  However, I am still having difficulties with the proof of Lemma 12. Unfortunately I don't have time to try hard to fix it but I feel there has to be at least some typos left that make it harder to understand....It also seems that the result is still not correctly stated.
> >
> > **Are there $log(n)$ terms appearing in the final bound or not?** In other words, is your $\mathcal{O}$ notation equivalent to $\widetilde{O}$, in that it allows for log terms?
> >
> >
> > In Lemma 11, I feel like the $log(n)$ terms disappear at the end of the first paragraph of page 17.  This seems to indicate no log terms. I don't really get how you get the extra additive term of $C^{(l)}$ at the second equation of the page...Is this the mean of the subgaussian variables that you need to withdraw as per Corollary 9? It seems so, but it should be done more explicitly.  In any case, assuming that the second equation there is correct, I agree that the log term disappears because the additive term $C^{(l)}$ dominates the RHS (this still requires n to be $polylog(\delta_1)$.
> >
> > **In Lemma 12 however things are more complicated: In equation (28), you definitely have extra log factors of n that you cannot get rid of. This is not stated in the statement of Lemma 12, and the proof is still  not thorough enough to make it unambiguous. Note that you do mention the polylog factors later in the statement and proof of Lemma 13 (but not in the main theorems of the paper)**
> > Also, you use the same notation $\delta_{in}$ AFTER the union bound when the $\delta_{in}$ no longer depends on $i$... This is all quite sloppy to the point of causing difficulties in the reading.
> >
> >
> > a couple of extra typos in the revision:
> >
> > page 17 (near top) : $\delta  1$ instead of $\delta_1$
> >
> > Page 17: "we have that with probability at least ..... that"
> >
> > page 17 (middle): again a $\delta  1$ instead of $\delta_1$
> >
> > page 21 (top): "we can simply the analysis"
> >
> >
> >
> >
> > 2. Thanks for the clarification regarding $O$. I originally kind of missed the fact that you treated the last layer so completely differently.
> > It seems like Lemma 14 is actually the key part of your argument then. Unfortunately I didn't have time to check all the details yet. You do say in the rebuttal that you "do not consider the case where O grows with n".  Do you think it would be difficult?
> >
> > 3. **Thank you so much for the clarification regarding the reference [1] in your rebuttal. It looks interesting and better explains where your work is placed in the literature. I think you need to explain it more clearly in the main paper though.** I have a clearer idea now and feel like your work is novel, though it is borderline for such a conference as ICLR (the original result in [1] is still the main point). Maybe adding more details including non asymptotic results with more explicit constants and a more explicit dependence on $O$ would make the work cross the (high) threshold for exceptional conferences such as ICLR, ICML etc.?
> >
> > I am increasing my score to 5 (it is definitely still below the threshold for ICLR in its current form), but I really would like to have a revision with a better proof of lemmas 12 and 13 (also a better positioning of the work within the literature and in relation to [1] in the main paper) . I hope I have time to look at the proof of lemma 14 in more detail at some point but I cannot guarantee it.

---

> > > ### Comment · Reviewer_bBN3 · 2022-11-18
> > > **error bars**
> > >
> > > Quick question (forgot to mention before).
> > >
> > > Why did you use error bars if there are indeed only three simulations? Wouldn't it make more sense to put the three observations there explicitly? Three observations is not even enough to determine the 25 percent quantiles which are used to plot error bars!

---

> > > ### Author Response · Authors · 2022-11-19
> > > **Response to Reviewer**
> > >
> > > Dear reviewer,
> > >
> > > Once again, we would like to thank you for very detailed feedback and analysis, which will surely help us improve the paper. In what follows, we have answered your questions and addressed your concerns. If any concern remains, we would like to continue the discussion, if time and rules permit. We have accordingly uploaded a new revision that you can download and check.
> > >
> > > > Are there  terms appearing in the final bound or not? In other words, is your  notation equivalent to
> > > , in that it allows for log terms?
> > >
> > > Indeed we meant to incorporate log terms into the $C$ terms. We have made this explicit in the new revision for all lemmas and tried to be as clear as we could be.
> > >
> > > >  In Lemma 12 however things are more complicated: In equation (28), you definitely have extra log factors of n that your ...
> > >
> > > Similarly, we have addressed the $\log n$ issue in this lemma as well, as stated before, and made all dependencies on log factors explicit.
> > >
> > > > Also, you use the same notation \delta_{in} AFTER the union bound when the  no longer depend
> > >
> > > We think that this is a confusion caused by notation. By $\delta_{in}$, we meant to point to the input probability argument to the lemma, not that $\delta$ depends on $i$ and $n$. We have changed this notation to $\delta_p$ to avoid such confusion in the newly updated revision. If we are wrong and the concern still remains, please let us know so that we can address your concern.
> > >
> > > > Thanks for the clarification regarding . I originally kind of missed the fact that you treated the last layer so completely differently. It seems like...
> > >
> > > Indeed Lemma 14 is the piece connecting the pNTK and the eNTK together, and deriving a bound on the difference of them. Previous lemmas capture the properties of eNTK, which also directly determine pNTK's properties as the two are closely related. These properties are then used in Lemma 14 to derive the required bounds. For the case where $O$ grows with $n$, we initially did not consider it as traditionally in the NTK literature the main focus is on supervised classification tasks, where $O$ is fixed (in fact to the best of our knowledge most related papers in the literature only consider the scalar output case). For our proofs, supporting the case where $O$ is not fixed would indeed introduce some difficulties in Lemma 14, since the shape of the final eNTK would also change with $O$. Hence, the convergence rate could slightly change (or at least new arguments would be required to show the same convergence rate in the case where $O$ grows with $n$).
> > >
> > > > Thank you so much for the clarification regarding the reference [1] in your rebuttal. It looks interesting and better explains where your work is placed in the literature. I think you need to explain it more clearly in the main paper though...
> > >
> > > If we understand correctly, by [1] you mean "On Exact Computation with an Infinitely Wide Neural Net - Arora et al 2019", which helps us derive a bound (although, assuming that their result extends to the multi-class case, which is at this point not clear to us) that is much slower than the bound presented here. We definitely agree that such a result should be mentioned at the related section and discussed more in the main body. However, we didn't know that we can indeed derive such a bound from that paper, and it was through your suggestion that we figured out that such a bound (again, in the case where their result extends to the multi-class setting) exists, and thus, we have not mentioned it in the related works section. Unfortunately, we still haven't had enough time to incorporate this in the main body, but we will do so as soon as possible.
> > >
> > > > Why did you use error bars if there are indeed only three simulations?
> > >
> > > Considering that the provided figures in the main body were already pretty crowded, we decided to include bars instead of putting dots for other seeds to make the figures readable as much as we could. However, we are already running the same experiments with new seeds and we will update the error bars accordingly when the new results are ready.
> > >
> > > Once again, we would like to thank you for your thorough and detailed review, which certainly helps improve the paper. We would also like to thank you for recognizing the novelty of this work. As we mentioned before, we believe that this work can be significantly impactful through easing the process of analyzing a network through the lens of NTK, while also providing new insights about the eNTKs of different networks at initialization and during training.
> > >
> > > Thanks,
> > > Authors

---

### Author Response · Authors · 2022-11-19
**General comment to all reviewers**

Dear reviewers,

We have uploaded a new revision of the paper in which we addressed some of the concerns that were shared among multiple reviewers:

* We have added new experiments supporting the theoretical results presented in the paper with other nonlinearities: GeLU, Tanh. And also, other initializations: Truncated Normal Initialization.
* We have changed the assumption of Gaussian initialization to any sub-Gaussian initialization method (such as uniform on an interval, truncated normal, etc), and accordingly changed all the proofs in the Appendix.

Please find the revisions in the newly uploaded version, and the experiments in section B.5 of the Appendix.

---

### Decision · Program_Chairs · 2023-01-20

**Decision:**

Reject

**Justification For Why Not Higher Score:**

The amount of revisions since the first submission is too large for if the paper were to have higher score (already bumped by one reviewer) to remain fair compared with other papers.

**Justification For Why Not Lower Score:**

Interesting topic, with contributions from the statistical and algorithmic field.

**Metareview: Summary, Strengths And Weaknesses:**

This paper introduces pseudo-NTK (pseudo-Neural Tangent Kernel), a provably efficient technique to estimate empirical NTK. The authors in particular show that the relative difference in Frobenius norm and maximum eigenvalue between the proposed approximation and the true empirical NTK is of order  $O(n^{−1/2})$ for deep networks of width n. Empirical results are provided that show the practicality of the results.

+ the paper looks at a deep learning model that is compatible with i) a connection with kernel methods and ii) is amenable to a theoretical analysis; it gives well-grounded insights on deep learning models;
+ the authors prove computational gains wrt to existing eNTK;
+ there are empirical results that are provided to test the efficacy of the method;

- if, as stated by the authors, the paper should be "looked at whole", the amount of revisions and changes done by the authors since the first submission plays a role; here, this amount is pretty big, containing correction of unclear statements, refactoring of some proof, new explanations; the paper is significantly different (and better) than the submitted version.